# Estimating small area population from health intervention campaign surveys and partially observed settlement data

Chibuzor Christopher Nnanatu [1,2] ✉, Amy Bonnie[1], Josiah Joseph[3], Ortis Yankey [1], Duygu Cihan[1], Assane Gadiaga[1], Hal Voepel [1], Thomas Abbott[1], Heather R. Chamberlain [1], Mercedita Tia[4], Marielle Sander[4], Justin Davis[5], Attila N. Lazar [1] & Andrew J. Tatem [1]

Effective governance requires timely and reliable small area population counts. Geospatial modelling approaches which utilise bespoke microcensus surveys and satellite-derived settlement maps and other spatial datasets have been developed to fill population data gaps in countries where censuses are outdated and incomplete. However, logistics and costs of microcensus surveys and tree canopy or cloud cover obscuring settlements in satellite images limit its wider applications in tropical rural settings. Here, we present a two-step Bayesian hierarchical modelling approach that can integrate routinely collected health intervention campaign data and partially observed settlement data to produce reliable small area population estimates. Reductions in relative error rates were 32–73% in a simulation study, and ~32% when applied to malaria survey data in Papua New Guinea. The results highlight the value of demographic data routinely collected through health intervention campaigns or household surveys for improving small area population estimates, and how biases introduced by satellite data limitations can be overcome.

Accurate small area data on human populations are vital for effective and equitable resource allocation and strategic planning. This includes responding to natural disasters[1], planning vaccination campaigns[2,3], and planning new schools[4]. While advances in the use of registers and administrative sources are opening up new possibilities[5,6], the primary source of small area data on populations in most countries remains national population and housing censuses[7]. The implementation of a census represents the largest peace-time operation that governments conduct, and as such, are typically undertaken only every 10 years. With population distributions and characteristics changing rapidly in many settings, census data can become quickly outdated, however[1]. Moreover, the huge resources required to conduct a census and the presence of inaccessible or conflict-affected areas, has meant that in many countries

full national enumeration has not been undertaken for more than a decade[7].

The substantial expense and logistical challenges inherent in conducting a census, as well as the difficulties in producing intercensal estimates at small area scales, have prompted the development of new approaches[8]. Principal among these recently are geospatial modelling approaches that draw on satellite imagery and small area sample 'microcensus' surveys to predict population distributions across countries[9]. These have included the development of modelled population estimates in Nigeria[10], Democratic Republic of the Congo[11], and South Sudan[12] based on bespoke microcensus enumeration data collection. While in each case, the production of modelled estimates proved cheaper and more rapid than conducting a full census, bespoke microcensus surveys can however still be complex logistically and

[1]WorldPop Research Group, School of Geography and Environmental Science, University of Southampton, Southampton, UK. [2]Department of Statistics, Nnamdi Azikiwe University, Awka, Nigeria. [3]National Statistical Office, Port Moresby, Papua New Guinea. [4]United Nations Population Fund, Port Moresby, Papua New Guinea. [5]Planet Labs, San Francisco, USA. ✉e-mail: cc.nnanatu@soton.ac.uk

expensive. Additionally, satellite-derived mapping of settlements can often be incomplete in tropical rural areas where tree canopies and cloud cover can obscure them[13]. These factors limit the wider application of geospatial modelling approaches to reliably address small area population data gaps.

Routine data collection from sources such as household surveys and health campaigns offer the possibility for obtaining recent population data that can form the basis of geospatial modelling approaches[14–16]. National household surveys, such as the Demographic and Health Surveys programme[17], typically collect household listings from sample locations before implementation of the full surveys. Moreover, health campaigns focussed on the widespread delivery of interventions such as vaccinations or bednets, often collect counts of numbers of people in households or in receipt of the interventions. These can represent a valuable source of geolocated, recent and reliable sample data on population counts, obtained with little additional expense compared to relatively costly bespoke microcensus surveys.

Another important input to geospatial population modelling approaches are reliable maps of building footprints or the extents of human settlements. Satellite imagery has been shown to be a valuable source of such data. Through manual delineation or machine learning algorithms, the extraction of these features across national and continental scales has been shown to be feasible[18,19]. However, the consistency and completeness of outputs can vary substantially[20], and a key limitation observed has been the challenge in detecting buildings and settlements under tree canopy cover[21]. The use of such data for mapping populations therefore brings a risk of substantial systematic bias in heavily forested regions.

Here we present and test a Bayesian statistical modelling approach that can utilise survey data collected routinely from health intervention campaigns or surveys, as well as adjusting for systematic biases caused by partially observed settlement data. The proposed method, herein, known as Two-Step Bayesian hierarchical modelling (TSBHM), corrects for bias in settlement data in the first step, and then uses the bias adjusted settlement data to predict population density at small area units in the second step.

We demonstrate the implementation and performance of the proposed approach using both a simulation study and real data application. In the simulation study, biases were implied by allowing proportions of settlements and survey data to be missing at random under different scenarios. We tested how the proposed model was able to recover the 'true' population from a biased sample and compared its estimation accuracy and error reduction rates against that of a more standard Bayesian Hierarchical Modelling (BHM) approach. We then implemented the approach for Papua New Guinea, a country lacking recent census data at the time of writing, and where rural settlements can be obscured by tree canopy cover. Model-based small area population estimates were produced using enumeration surveys undertaken for malaria bednet delivery with both the BHM and TSBHM approaches.

## Results

### Simulation studies

Results of the simulation studies showed that the BHM and TSBHM approaches were more sensitive, with a clear trend, to the changes in the proportion of observed satellite-based settlement data than to the changes in the proportion of census units with observed population counts. These impacted more on the estimates based on the BHM method than the estimates of population counts obtained using the TSBHM approach.

In Fig. 1, across all the model fit metrics, both models performed unsurprisingly equally well at 100% survey coverage and 100% settlement observation coverage. This indicates that any one of the BHM and TSBHM approaches could be used when there is full enumeration, and settlements were perfectly observed. However, with partially

observed settlements, the performance of the BHM method deteriorated much more rapidly than the TSBHM method, which remained superior in performance and only gradually waned as the proportion of unobserved settlements increased (see also Fig. S1 of the supplemental document for more model fit performance check graphs).

In Fig. 1A, B, the TSBHM method yielded lower root mean square error (RMSE) with higher correlation coefficient (CC) statistics. Similarly, in Fig. 1C, we compared the accuracy of the population predictions based on the two models at 100% survey coverage. More accurate estimates of population were provided by the TSBHM approach with relatively narrower 95% credible intervals in Fig. 1C. Also, the violin plots in Fig. 1D showed that the TSBHM-based estimates were more accurate and less uncertain than the BHM approach across all satellite observation coverage proportions. Note that the missingness with respect to survey coverage is assumed to be either missing completely at random or just missing at random. A missed observation is said to be missing completely at random (MCAR) if it is only missed at random sets of locations; while missing at random (MAR) could normally involve missed observations at clusters of locations due to some shared characteristics (e.g., [22]).

Model performance checks based on reduction in relative error rates showed that the TSBHM approach consistently outperformed the BHM approach across various combinations of proportions of observed settlements and survey coverages. Specifically, Fig. 2 shows that the TSBHM approach caused 32% − 73% reduction in relative error rates over the traditional BHM approach. However, the amount of reduction in relative error rates decreased gradually as the proportion of missing settlement information increased (see also, Fig. S2 of the supplemental document).

### Papua New Guinea application

When implemented to produce small area population estimates using enumeration surveys across Papua New Guinea, results showed that the TSBHM approach improved estimation accuracy and produced reductions in relative error rates relative to the BHM approach. Model fit checks and cross validation results based on the best fit BHM and TSBHM population density models are presented in Table 1, while the posterior estimates of the model parameters are presented in Table S1 of the supplemental materials.

Note that the predicted building intensity in step 1 of the TSBHM approach, which was the input settlement information used in step 2, was based on the best fit models. Preliminary models which assessed the potential impacts of the differences in the data collection strategies employed by the various data sources by including a data source random effect (*source*) in the model, showed that the BHM-based model without the data source random effect produced lower DIC values than the one which accounted for data source differences (−217,407.9 versus −216,071.1). Thus, the data source random effect was dropped for the TSBHM-based models to enable direct comparison between the BHM best fit model and the corresponding TSBHM-based model. However, for ease of exposition, only the posterior estimates based on the best population density models are discussed. In both tables (Table 1 and Table S1), model performances and posterior parameter estimates were compared across the BHM and TSBHM methods. Again, from Table 1, the TSBHM method provided consistently better fit and cross-validation metrics than the BHM based models. In addition, both in-sample and k-fold out-of-sample (with $k = 5$) cross-validation results based on the population 'density' model indicates that the TSBHM model performed substantially better, with lower MAE, lower RMSE, and lower absolute bias, even though the correlation coefficients for both methods are approximately equal at >86%.

In Table S1, the posterior parameter estimates of the best fit models for the BHM and TSBHM methods are presented. Of the 15 geospatial covariates, the standard deviation of the estimates based on

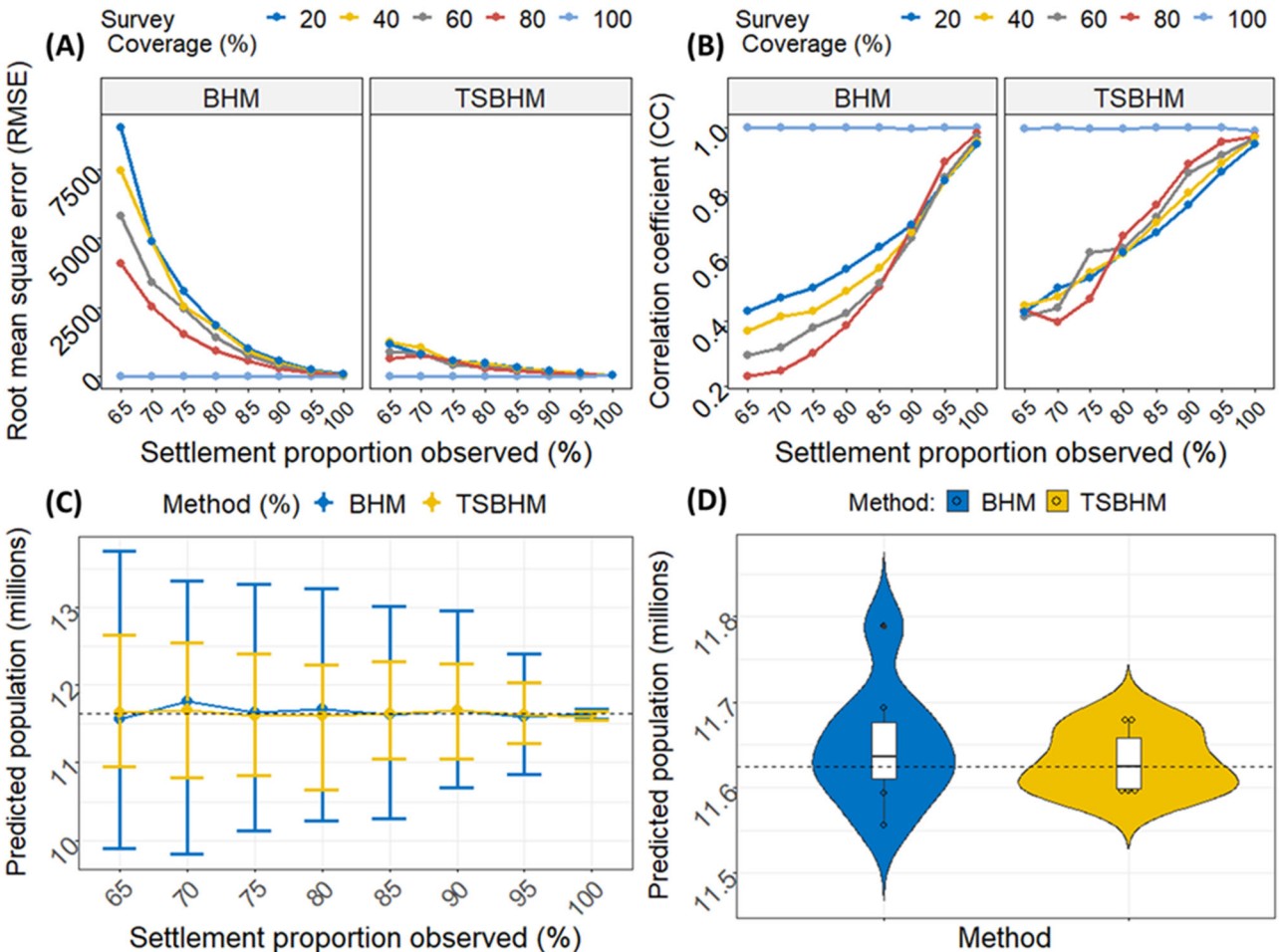

**Fig. 1 | Model fit metrics graphs based on the simulation study.** The simulation study provides an opportunity to evaluate the performance of the proposed method against the conventional approach. Across the model fit metrics used, the TSBHM method consistently provided better model fits than the BHM approach. **A** Root Mean Square Error (RMSE), **B** Correlation Coefficient (CC) between the observed and predicted counts, **C** Error bar plots with 95% credible intervals and, **D** Violin plots of the predicted national total population across various proportions of satellite observation coverages using both BHM (blue) and TSBHM (orange) methods, when population data are 100% observed.

the TSBHM method were consistently lower, indicating higher estimation accuracy. The results identified 'distance to main road (cov7)', 'Distance to shrub area edges (cov16)' and the 'slope (cov20)' as the 3 top variables that positively influenced the spatial distribution of population density across the 32,100 CUs in PNG. However, the largest negative correlation occurred with 'distance to marketplace (cov8)'. These results suggest that, unsurprisingly, the majority of the PNG population live closer to marketplaces in rural areas that are far from major highways. Additionally, Figure S3 of the supplemental document shows the nicely mixing trace plots of the posterior distribution of the population estimates of 6 randomly selected census units based on 2000 posterior samples after taking 20% as burn-in. This shows that the use of the INLA-SPDE approach ensures that the parameter estimates are obtained from the stationary distribution of the target density with no convergence issues.

Furthermore, we used the scatter plots in Fig. 3 to assess the correlation between the observed population and the predicted population counts based on the best fit model of the TSBHM approach at both census unit level (Fig. 3A) and at the district level (Fig. 3B).

Results indicate that there is an agreement between the observed and predicted population counts at both census unit and district levels with lower correlation obtained at the district level. The lower correlation at district (higher) level reflects the fact that the observed district level aggregated counts contain a much higher proportion of missing census unit counts. Additionally, results show that the TSBHM model caused a 32% reduction in relative error rates over the BHM approach.

Figure 4 shows a bar chart of the province level population estimates along with error bars representing the 95% credible intervals based on the BHM and TSBHM methods. Although both methods produced predictions with similar trends, the BHM appears to produce higher and more uncertain estimates of population for most provinces, except for New Ireland, Northern (ORO) and North Bougainville, where very similar estimates were produced by both methods. Moreover, the consistently shorter width of the 95% credible interval for the TSBHM method indicates a higher level of prediction accuracy.

Additionally, the spatial distribution of the census unit (CU) level population counts estimates obtained using the TSBHM method, as well as the estimates of uncertainty, measured through the difference between the upper and lower bounds of the 95% credible interval divided by the mean (that is, (upper – lower)/mean), are shown in Fig. 5. Specifically, Fig. 5A shows the spatial distribution of the predicted population counts at the CU level, while Fig. 5B shows the spatial distribution of the estimates of uncertainty in the posterior estimates of population, also at the CU level. Smaller values of uncertainty are indicative of a higher predictive accuracy, thus, in Fig. 5B, the areas with lower values imply areas of higher accuracy in the predicted CU-level population counts based on the TSBHM

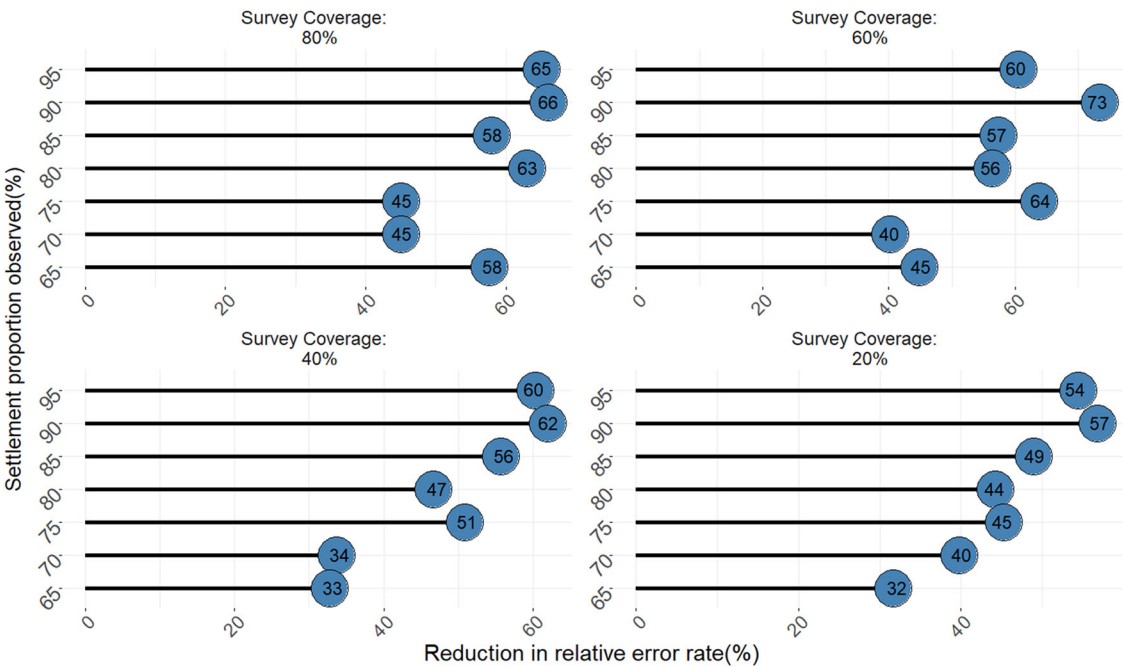

**Fig. 2 | Reductions in relative error rates produced by the proposed method against the conventional method.** The numbers in the blue circles of the lollipop graphs are the percentage reductions in relative error rates produced by the TSBHM approach over the conventional BHM approach based on the simulation study across various proportions of observed settlement and survey coverages. Percentage reductions in the relative error rates decreased as the proportions of missing settlement data increased and ranged from 32%–73%.

method. Estimates of uncertainty varied in space, and areas with little or no data are characterised by higher uncertainty.

## Discussion

Geospatial modelling methods built using data from satellite imagery have been shown in multiple settings to provide a mechanism for filling demographic data gaps[10,11,23–25]. Differences in national needs, situations and data types, however, mean that a one-size-fits-all model could rarely be the best option. The need for a 'toolbox' of model types to tackle different national scenarios and situations is therefore clear. Previous approaches have been designed for situations where disaggregation of areal estimates to grid squares is required[26] or where predictions are needed from incomplete censuses[23] or bespoke microcensus surveys[11]. Here, we have developed, tested and applied a new approach to add to the national population modelling toolbox in situations where health campaign data are available and satellite-derived settlement data are incomplete. The approach, which was

based on Bayesian statistical inference framework, was implemented via the integrated nested Laplace approximation in conjunction with stochastic partial differential equation (INLA-SPDE[27,28]) strategies. Therefore, it allowed us to easily quantify uncertainties in the model parameter estimates using the 95% credible intervals of the posterior distributions. Moreover, the model outputs (including the posterior distributions) can serve to support governance, reporting and tracking systems used by governments and international agencies[1,12], and are already in use for country planning in Papua New Guinea[29].

The simulation study showed how different modelling approaches and levels of data completeness impacted population prediction accuracies and demonstrated the value of the TSBMH approach introduced in this paper. The results provide valuable insights into the generalisability of the methods tested to a range of different situations that may be faced across the world. The validation statistics show that when both the population data and settlement building data are fully observed across the entire spatial units of interest, both the traditional BHM approach and the proposed TSBHM approach performed well. However, when only a proportion of the settlements could be observed (due to, for example, cloud or tree canopy covers), the TSBHM approach consistently provided better cross-validation (predictive) performance than the traditional BHM method. Specifically, the TSBHM approach provided better predictive performance in the out-of-sample cross-validation with lower mean absolute error (MAE; 0.007 versus 0.67), lower root mean square error (RMSE; 0.176 versus 14.716), and smaller absolute bias (Abias; 0.005 versus 0.665). Moreover, at the national level, the TSBHM approach showed -32–73% reduction in relative bias across different combinations of the proportion of the observed population count versus the proportion settlements.

While the simulation study was a valuable test of different modelling approaches and their generalisability, it was important to also implement the methods in a real world setting to demonstrate their applicability. PNG provided an ideal test case, given the availability of recent partial enumeration data from a health campaign, and evidence of rural settlements under canopies that were obscured in satellite

**Table 1 | Model fit metrics based on k-fold cross-validation using observed datasets on Papua New Guinea**

| Metric | In-Sample Cross-Validation | | | | Out-of-Sample Cross-Validation | |
|---|---|---|---|---|---|---|
| | BHM (Count) | TSBHM (Count) | BHM (Density) | TSBHM (Density) | BHM (Density) | TSBHM (Density) |
| MAE | 3.191 | 2.251 | 0.062 | 0.005 | 0.670 | 0.007 |
| RMSE | 5.734 | 4.213 | 1.891 | 0.220 | 14.716 | 0.172 |
| Abias | 3.142 | 2.123 | 0.054 | 0.003 | 0.665 | 0.003 |
| CC | 0.991 | 0.991 | 0.992 | 0.792 | 0.866 | 0.862 |

The predictive abilities of the proposed TSBHM and the conventional BHM methods were evaluated using a k-fold cross-validation technique. Model performances were evaluated and compared based on Mean Absolute Error (MAE), Root Mean Square Error (RMSE), Absolute bias (Abias), and the Pearson correlation coefficient (CC). Lower values of MAE, RMSE, Abias and higher values of CC indicated higher predictive ability.

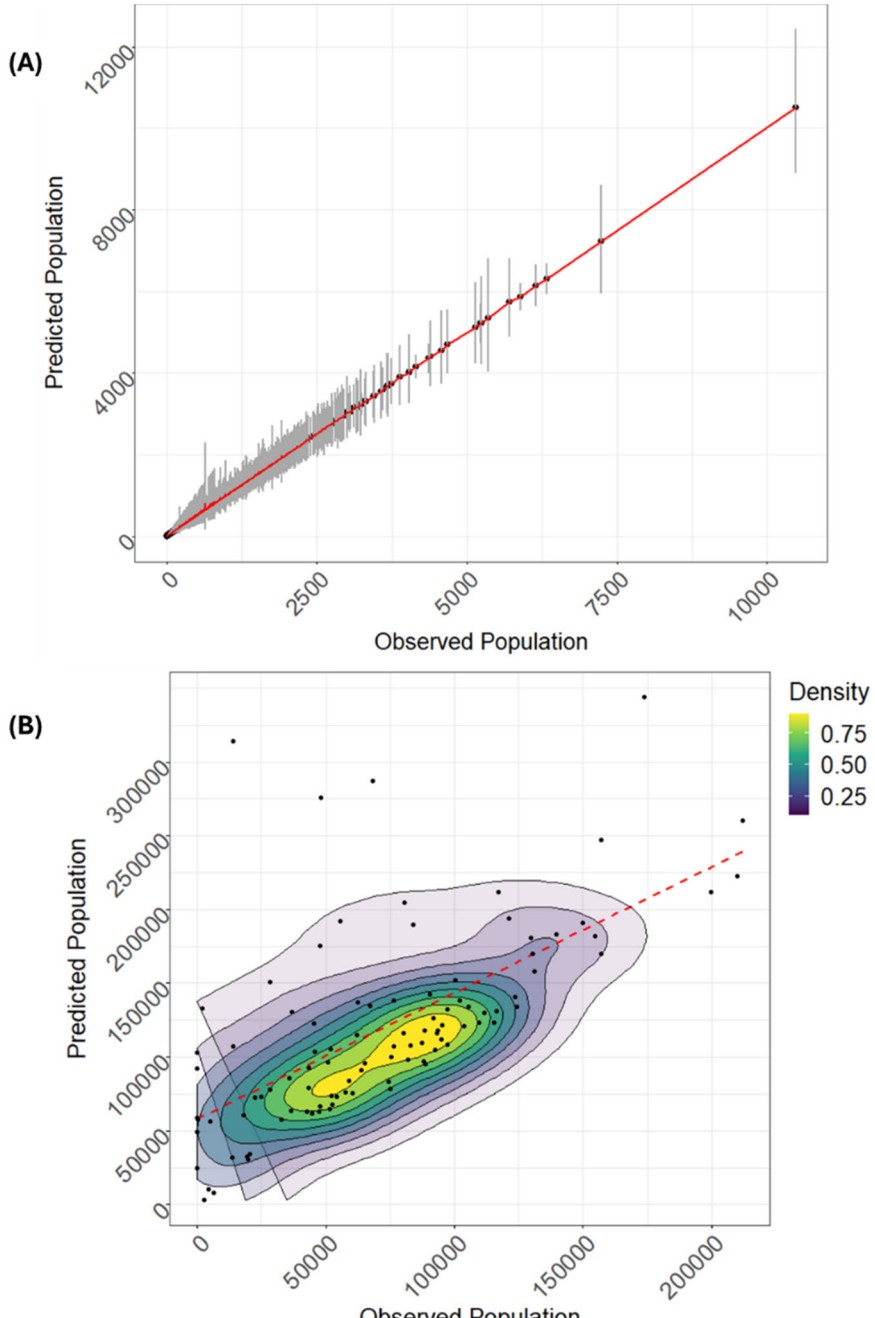

**Fig. 3 | Model fit assessment graphs for the Papua New Guinea application.** The correlation assessments between the observed data and the predicted population counts based on the proposed TSBHM approach at both the census unit and the district levels indicated good predictive ability of the model. **A** Scatter plot of observed counts vs TSBHM predicted counts with 95% credible intervals (vertical grey lines) across census units. **B** Density scatter plot with contour lines of observed versus TSBHM predicted counts at the district level.

imagery. Implementation of the BHM and TSBHM approaches showed that the proposed TSBHM approach caused ~32% reduction in relative bias at the national level, and put the total national population estimate at ~11.78 M (95% CI; lower = 11.64 M, upper = 12.03 M) people. The data outputs which were further disaggregated by age and sex groups provide important support to several decision-making processes and governance in PNG. These datasets are now publicly available on the websites of PNG's National Statistical Office (NSO)[29] and within WorldPop's data repository[30].

Moreover, the results of the simulation study and implementation highlight the value of the proposed new modelling approaches in the production of accurate small area population estimates in the absence of full enumeration. Nevertheless, limitations and uncertainties remain that should be noted and explored in future work. For example, when the proportion of the missing/unobserved settlement data increased significantly, uncertainties in the population estimates also increased regardless of whether the TSBHM or BHM approach was used. Although, within the TSBHM approach, the increase in uncertainties was only gradual while there was a rapid increase in the uncertainties of estimates for even a 5% increase in the proportion of missing settlement data when the BHM approach is being implemented. Additionally, although it is expected that the application of the TSBHM approach to a much higher resolution grid-cell level, say at 100 m, would be very straightforward, it is most likely to be computationally

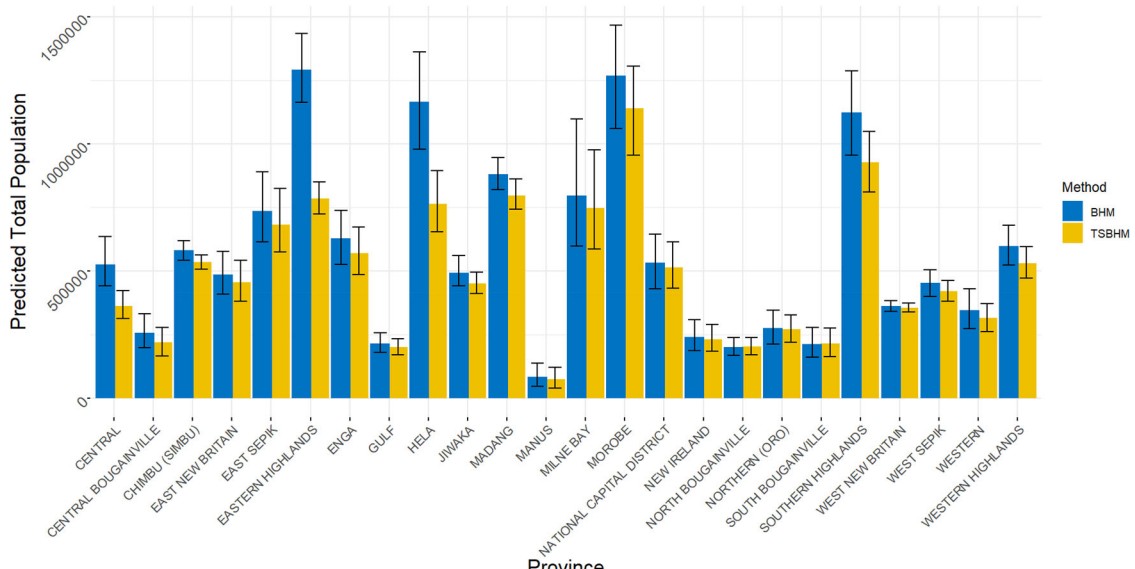

**Fig. 4 | Bar charts comparing the posterior population estimates produced by the BHM and TSBHM approaches across Papua New Guinea provinces.** Bar charts with 95% credible interval error bars (black) for comparing the posterior population estimates of Papua New Guinea provinces based on the BHM (blue) and TSBHM (yellow) approaches. The error bars provide a means of assessing the uncertainty in the parameter estimates as the distance between the predicted lower and upper bound population estimates based on the 95% credible intervals of the posterior distribution. Shorter error bars indicate higher accuracy. The TSBHM method consistently provided comparatively shorter error bars compared to the BHM approach, thus suggesting higher prediction accuracy.

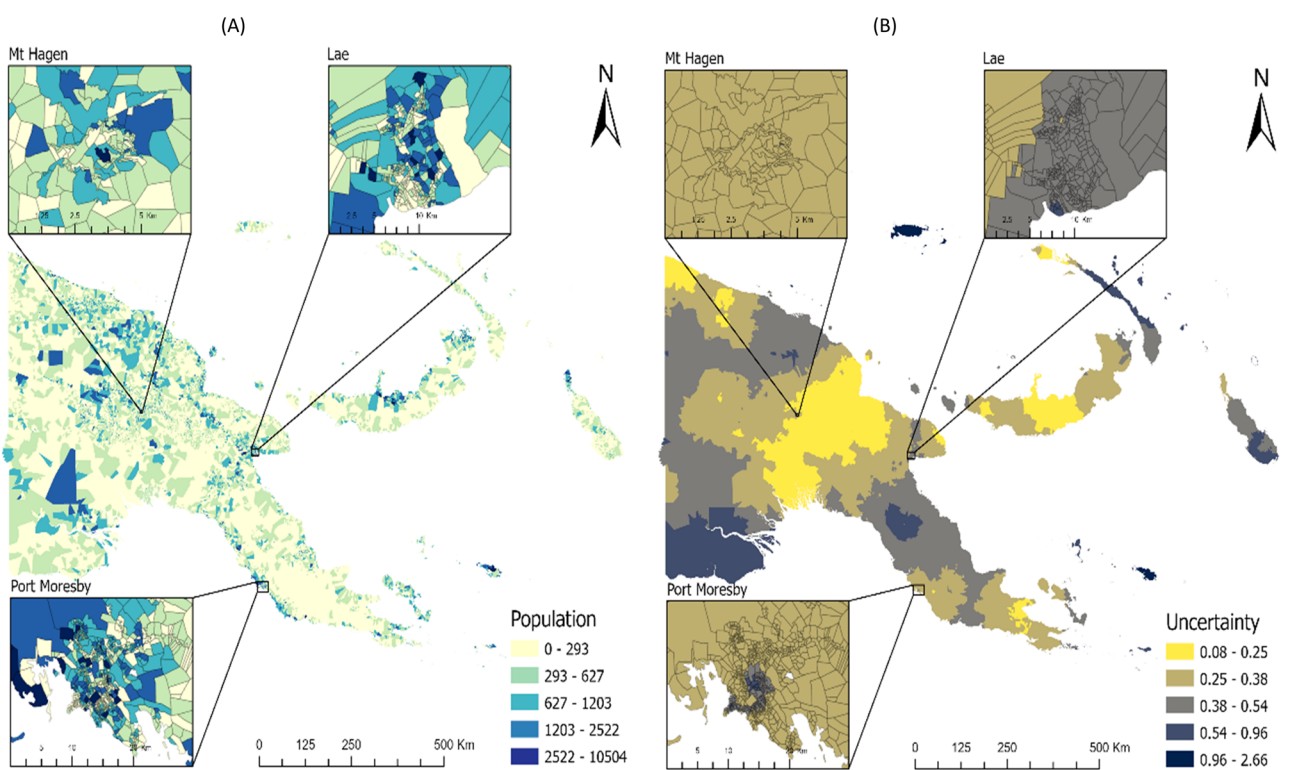

**Fig. 5 | Spatial distributions of the posterior predictions at the census unit (CU) level using the TSBHM approach. A** Posterior estimates of the population numbers and (**B**) the associated measures of uncertainty across the census units in Papua New Guinea. The uncertainty values were obtained by dividing the difference between the upper and lower bounds of the 95% credible intervals by the associated mean value. Smaller uncertainty values indicate higher prediction accuracy.

expensive, and this is the focus of an ongoing study. Also, within the current study, we have only utilised the posterior mean of the predicted building intensity obtained in step 1 to estimate population density in step 2. However, future work could test the integration of step 1's estimates of uncertainties of the building intensity into the estimation of population density in step 2 to explore whether this provides more robust estimates. Finally, it should be noted that while the sophistication of the approach in terms of ability to overcome data

limitations and provide reliable population estimates with uncertainty metrics is a major advantage, its complexity also represents a disadvantage. For such approaches to be adopted and used by governments, understanding and ownership of the methods and outputs is important. This requires strong skills in Bayesian statistical modelling to be present within national statistical offices, which can be strengthened through close collaborations with academic partners, capacity strengthening initiatives and regional support networks.

The findings and outputs outlined in this paper point towards a range of next steps in terms of advances in methods, data and applications. Advances in satellite and drone imagery, and abilities to extract valuable insights from them may reduce the need for methods to address the issues of settlement canopy coverage. High resolution synthetic aperture radar imagery can potentially detect buildings underneath tree canopies[31], while the mapping and AI-based extraction of buildings from multi-temporal and multi-view angle series of images could potentially improve abilities to detect structures when they become more visible[32]. Nevertheless, we anticipate that challenges could remain in mapping populations in certain types of terrains, such as mountainous, desert and snow-covered areas, where the accuracy of satellite-based human settlement mapping tends to be lower due to topographic variations and the similarities between human settlements and the surrounding landscapes[33,34].

The simulation study provides quantification of how well the methods perform under varying levels of population and settlement completeness and provides guidelines for implementation in other settings where similar demographic data challenges exist. One possibility is Democratic Republic of the Congo (DRC), where no national census has been performed since 1984, but multiple health campaigns are providing valuable recent enumeration data, and evidence of settlements obscured by canopies exists[13]. The study here highlights the value of routinely collected population enumeration data from campaigns and national surveys, and the possibility to leverage these to produce small area population estimates in settings where census data are outdated. Recent work in Zambia[15] and Cameroon[24] has shown the value of household listing data from surveys as a source of training and validation data for population models. The work here also emphasises the possibility of a 'virtuous circle', where health campaigns can make use of modelled small area population estimates to design their implementation, but then collect new enumeration data during this implementation to then update and improve modelled population estimates. Undertaking this requires strong partnerships between modelling teams, ministries of health, national statistical offices and any implementing partners.

## Methods

In this section, we provide details of the motivating datasets as well as the background of the proposed methodology. We begin with the data description followed by the statistical modelling techniques as well as the implementations through a simulation study and real data application.

### Data

Three sources of data were identified, analysed and processed to provide the basis for population estimation modelling in Papua New Guinea (PNG). These were (i) Population surveys containing full enumerations of sample units for providing model training and validation data; (ii) Settlement maps derived from satellite imagery to provide a key input in the prediction of population numbers and distributions; (iii) Geospatial covariates for testing and use in modelling population distributions.

### Population survey data

Two sources of recent geographically referenced population enumeration data were used for model training and validation. Each was chosen because they captured full enumerations of the lowest available administrative units (Census Units, CUs) and had extensive spatial coverage, ensuring good representation of urban and rural areas across the country. These were urban structural listings (USL) collected in 2021, and enumerations collected in 2019-21 to support the distribution of anti-malarial long-lasting insecticide-treated bednets. For the USL, 1,959 CUs were enumerated in 2021, and for the malaria bednet surveys, 15,468 CUs were enumerated in the 2019-21 period. After accounting for overlaps (where census units had enumerations from both sources of data), a total of 16,903 census units had survey data available for training the population model. The spatial distributions of the datasets are shown in Fig. 6.

The USL was carried out in 2021 by the National Statistical Office (NSO) as a prerequisite for the census[35]. The survey records the number of inhabitants within each structure, along with structure types and characteristics. The aim was to carry out the survey in all province capitals, major cities and some of the major district capitals. Due to funding constraints, not all urban areas could be covered, however the CUs that were surveyed were done so to full enumeration.

Rotarians Against Malaria (RAM) is a PNG based Non-Governmental Organisation (NGO) that works in collaboration with the National Department of Health (NDOH) along with other organisations such as UNICEF, WHO and church groups to reduce and control the impact of malaria in PNG. RAM's primary activity is the distribution of long-lasting insecticidal nets (LLINs) across PNG with each village visited once every 3 years. Population surveys are carried out routinely prior to net distribution to ensure every household gets adequate number of nets. Sometimes, it is not possible to visit all the villages in each round of survey due to factors such as villages that are too remote, shortage of funds, and other factors.

The USL (1959 CUs, 2021) and the most recent malaria surveys (15,468 CUs, 2019-21) were used in the population estimation model, covering ~52 percent of the land area of PNG.

### Settlement maps

High resolution data on the presence of buildings across PNG were obtained from Planet (www.planet.org) to map the distribution of human settlement. Planet's road and building segmentation model is a modified UNET, a deep learning supervised segmentation model, that runs on 4-band (red, green, blue and near infrared) surface reflectance imagery. It was trained on around 4000 Planetscope satellite images (https://developers.planet.com/docs/data/planetscope/) paired with hand-annotated labels to mark each pixel into one of 3 classes: road, building, or other. The model takes in Planetscope data and generates a 3-band raster result, where each pixel has a value of 0–255 for each band, and each band has an assigned class (road, building, or other). The higher the value, the more confident the model is that the pixel belongs to its class. Planet publishes imagery covering almost the entire landmass of the world, nearly every day. While the model was trained and runs on individual images, improved results can be obtained when the data density which PlanetScope provides is used.

For a given area, the segmentation model is run on all of the published PlanetScope data for a given week, month, or quarter and then those results are aggregated into a single result. This process removes any noisy detections caused by atmospheric interference, provides the greatest coverage and generally ensures the best possible representation of roads and buildings for the location. The Planet experimental non-public building and roads data product was downloaded from the Planet API, where it was made available to the authors through a sharing agreement with the University of Southampton as a tiled product. Further information on the data can be found at: https://www.planet.com/pulse/mapping-all-of-earths-roads-and-buildings-from-space/. The settlement data product provides information on the locations of buildings in gridded (raster) format. The output gridded dataset was created through the classification of available cloud-free

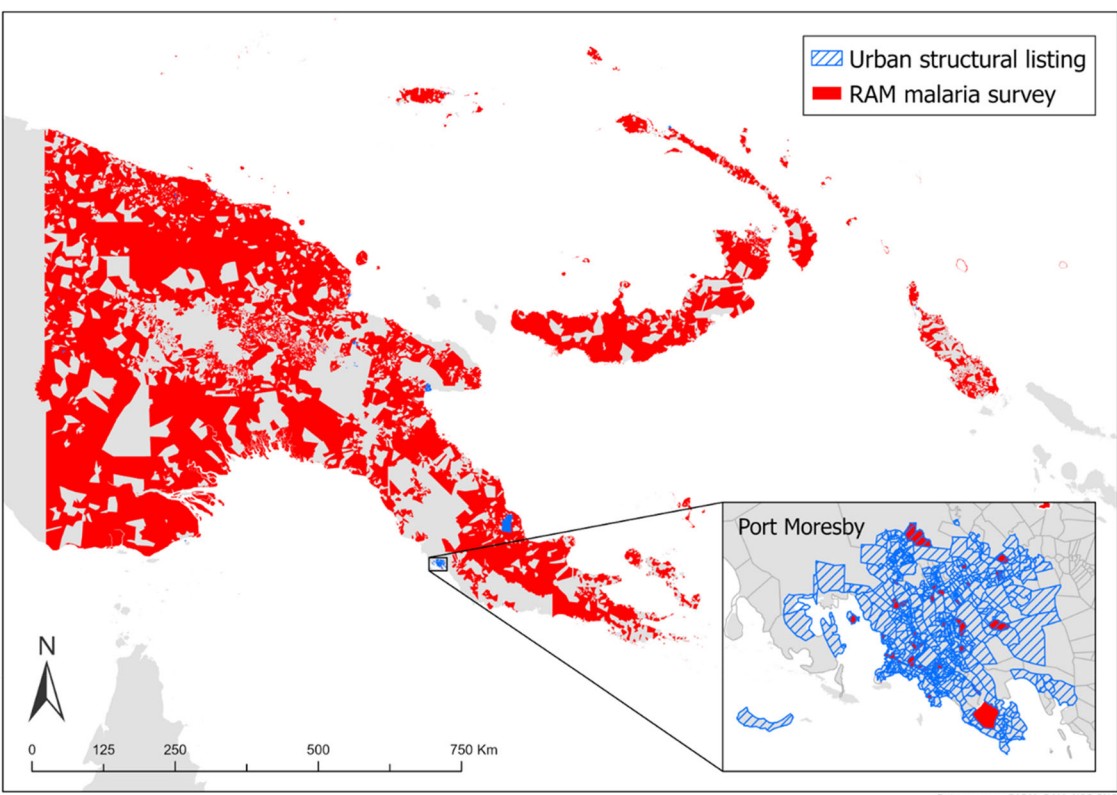

**Fig. 6 | Spatial distribution of the census units in Papua New Guinea.** The map shows the locations of the 16,903 census units across the country where recent population count datasets from the Rotarians Against Malaria (RAM) survey and urban structural listing are available, as well as where they overlap.

satellite imagery from a 7 month period (July 2021–January 2022), with classified outputs compiled to create a single gridded output. Values in the output dataset ranged from 0 to 254 with higher values indicating a higher likelihood of building/road presence.

The gridded dataset in geoTiff format had a spatial resolution of ~4.77 metres and was projected in World Mercator Auxillary Sphere (EPSG:3857). A common spatial resolution (3 arc seconds, ~100 m at the Equator) and grid cell alignment was defined to ensure spatial harmonisation of all input gridded datasets (settlement and geospatial covariates). The Planet settlement data was resampled from 4.77 m to 3 arc second resolution, with grid cell values summed in the resampling process to provide a settlement "intensity" measure. Prior to the resampling step, a value of 1 was subtracted from all raster values, such that all non-settled grid cells had a value of 1, and the maximum value was 253. The settlement intensity values in the resampled raster ranged from 0 to 88,237.

Given the known challenges of detecting human settlement under tree canopies outlined above, manual comparisons of the Planet data and recent high resolution satellite images against the sample population enumeration data were undertaken (see Fig. S4 of the supplemental materials for an example and further details). These comparisons highlighted forested rural areas where substantially larger numbers of people were enumerated compared to area of settlement and buildings mapped, confirming that settlements had likely been missed. Additional comparisons against other satellite-derived settlement mapping datasets (Global Human Settlement Layer (GHSL)[36], World Settlement Footprint (https://geoservice.dlr.de/web/maps/eoc:wsf2019)), found similar or worse issues in terms of missing settlement.

### Geospatial covariates
Based on findings from previous population modelling studies (for example[26,37]), a wide range of geospatial covariates were considered in the development of the modelled population estimates, including

those representing aspects of topography, climate, land cover, infrastructure and human settlement information. In total, 52 geospatial covariates were considered at the start of the covariate selection process (see Table S2 and Table S3 of the supplemental materials for details on the final sets of covariates selected within the best fit models). All geospatial covariates were created as gridded (raster) datasets with a harmonised 3 arc second (~100x100m) spatial resolution and grid cell alignment, from which spatially aggregated summary statistics were calculated.

### Statistical modelling
In this section, we provide an outline of the statistical models employed for this study. We outline how a Bayesian hierarchical model (BHM), implemented within a bottom-up population modelling context[11,23,24] can be extended to a two-step model (TSBHM) developed to make use of routinely collected demographic data from health campaigns and address the issue of partial observation in satellite-derived settlement data. We also outline details of a simulation study to examine how different model types perform with varying levels of missing data, before outlining the application to PNG using the demographic data described above. The workflow of the modelling framework is provided in Fig. 7 and shows how geospatial covariates are used to predict settlement data at unobserved locations in the first step, which were then used as model inputs to predict population density in the second step.

### Two-step Bayesian hierarchical modelling (TSBHM) framework for bottom-up population modelling with partially observed settlement data
The basics of the Bayesian hierarchical model (BHM), as implemented within a bottom-up population modelling context, are outlined within the 'Methods' section of the supplemental information (see also Table S4 of the supplemental for the description of

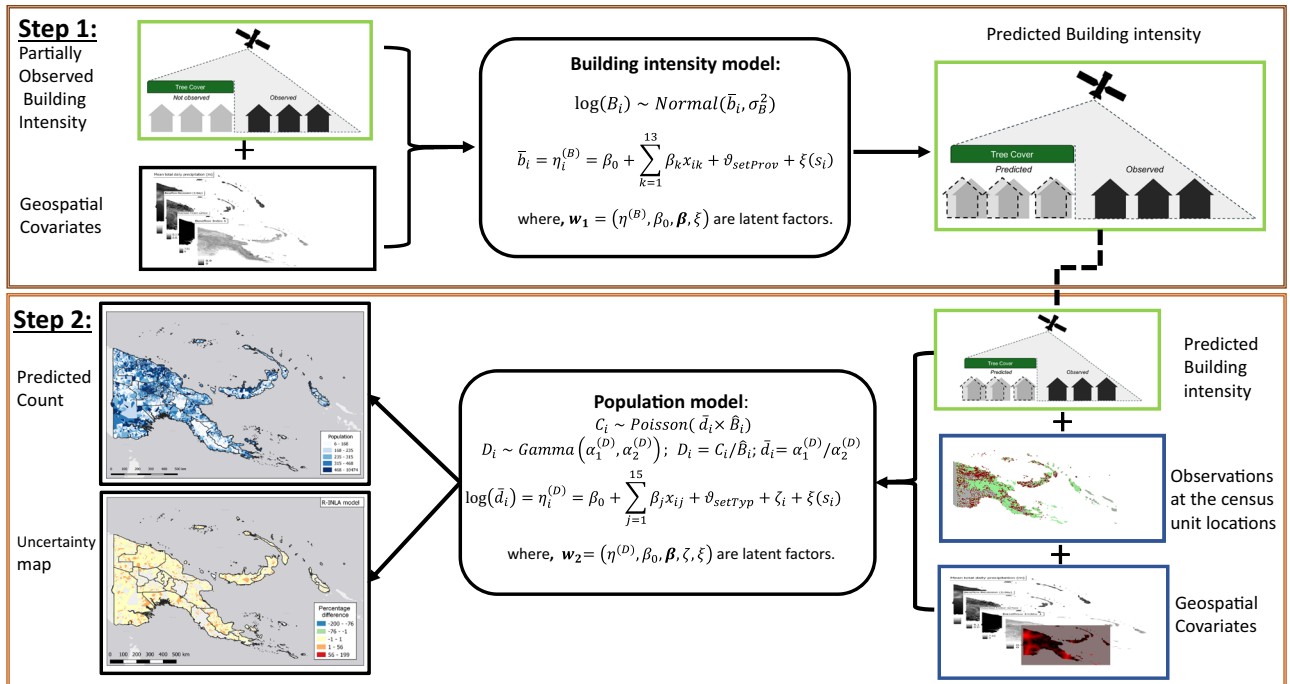

**Fig. 7 | Overview of the two-step hierarchical regression model workflow.** Biases in the observed settlement data are corrected in step 1, while population estimates are produced in step 2 using the bias-corrected settlement data from step 1. $C_i$ is the count of people in census unit $i$; $D_i$ and $B_i$ are the population density and building intensity for census unit i, respectively; $\widehat{B}_i$ in Step 2 is the predicted building intensity from Step 1; $\vartheta_{(\cdot)}$ are zero-mean Gaussian random intercepts for settlement types (*setTyp*) and settlement type versus Province interactions (*setProv*). The spatial random effect $\xi$ is modelled via Gaussian Markov random field (GMRF)[28].

mathematical symbols and notations used herein). As described above, the Planet settlement data were resampled, and grid cell values summed to provide a measure of settlement intensity (hereafter known as the *building intensity*). Please note that building intensity and building counts, although not the same, have been used interchangeably in this manuscript and in the supplementary document to serve as proxy to human settlements within the population models. The values in the resampled raster at 3 arc second spatial resolution, ranged from 0–88,237, with values of 0 representing no detected settlements and values increasing with area and certainty of detected settlement. The BHM scheme[9–11,23,24] assumes that the input settlement building intensity $B_i$ is perfectly observed (that is, unbiased). However, this is not likely to be the case in those rural villages where settlement structures are obscured by tree canopy covers. In this case it makes sense to assume that $B_i$ is not perfectly observed. Unrealistically small values of $B_i$ would lead to inflated estimates and inaccurate predictions (see, for example, Fig. S5 of the supplemental materials). To circumvent this, here, we introduce a two-step modelling solution based on the BHM framework described in the earlier section, henceforth known as TSBHM, which first adjusts for the imperfections within the building intensity data $B_i$, and then uses the predicted (bias-adjusted) building intensity $\widehat{B}_i$ as a model input for estimating the population count in step two.

### Step One

In step one, the response variable is the imperfectly observed building intensity $B_i$ whose natural logarithm is assumed to be normally distributed with mean $\bar{b}_i$ and variance $\sigma_B^2 > 0$. That is,

$$\ln(B_i) \sim \text{Normal}(\bar{b}_i, \sigma_B^2) \qquad (1)$$

We let the expected value of the building intensity $\bar{b}_i$ relate with a set of geospatial covariates $x_1,\ldots,x_K$ and other auxiliary variables $z_1,\ldots,z_L$ through the identity link function $g(\bar{b}_i)$ given by $g(\bar{b}_i) = X_i\boldsymbol{\beta} + Z_i\boldsymbol{\vartheta} + \zeta_i + \xi(s_i)$, where $X_i$ and $Z\_i$ are the design matrices of

the geospatial covariates and the auxiliary variables including random effects such as settlement types (*setTyp*), Province (*prov*), and their interactions (*setProv*); $\boldsymbol{\beta}$ and $\boldsymbol{\vartheta}$ are vectors of unknown regression and random effects parameters; and $\zeta_i$ and $\xi(s_i)$ are the spatially independent and spatially varying random effects corresponding to location $s_i$, respectively. Note that the building intensity variable is discrete and could also be modelled using discrete count probability distribution such as Poisson distribution. However, for our purposes, we have used normal distribution to model the log-transformed version of the data which was continuous and normally distributed (based on preliminary explorations). Note also, that here, we use interchangeably both the variance parameter $\sigma_B^2$ and the corresponding precision parameter $\tau_B = 1/\sigma_B^2$. Note that apart from the normal distribution, any other skewed distribution (including skewed-normal, log-logistic, and Gamma densities) could be used to model the logarithm of the building intensity. However, the use of the normal distribution appears to work best in our context.

The full hierarchical model for predicting the bias-adjusted building intensity is given by

$$\ln(B_i) \sim \text{Normal}(\bar{b}_i, 1/\tau_B)$$

$$g(\bar{b}_i) = \eta_i^{(B)}$$

$$\eta_i^{(B)} = \beta_0 + \sum_{k=1}^{K} \beta_k x_{ik} + \sum_{m=1}^{M} \boldsymbol{A}_{im}\xi_m + \vartheta_{setTyp} + \vartheta_{Prov} + \vartheta_{setProv} + \zeta_i$$

$$\pi(\beta_0) \propto 1$$

$$\beta_k \sim \text{Normal}(\mu_\beta, 1/\tau_\beta)$$

$$\vartheta_l \sim Normal\left(0, \frac{1}{\tau_l}\right); l \in \{setTyp, Prov, setProv\}$$

$$\xi \sim SPDE$$

$$\zeta_i \sim Normal(0, 1/\tau_e)$$

$$\tau_j \sim Gamma(\alpha_1^{(j)}, \alpha_2^{(j)}), where\ j \in \{B, \beta, l, e\} \qquad (2)$$

where $\eta_i^{(B)}, \beta_0, \beta_1, \ldots, \beta_K$ are the structured additive linear predictor, the intercept parameter, and the unknown regression coefficients of the $K$ geospatial covariates, respectively; $\vartheta_l$ is the Gaussian distributed random effect term which accounts for variabilities in the observed building intensity due to latent effects of settlement type, Province or the interactions between settlement types and Province. For PNG, there are three settlement types (urban, rural and non-village) and 24 Provinces which constitute $3 \times 24$ (=72) permutations of settlement type versus Province interactions; $A$, $\xi_m$ and $\zeta_i$ are the projection matrix which maps the observed building intensities to the mesh nodes (see Fig. S7 of the supplemental materials for the mesh with $M$(=900) nodes used for the models), the spatial random effects implemented as sparse Gaussian weights through the SPDE framework[28], and the zero-mean Gaussian distributed spatially independent random effects to account for any other variabilities that remain unexplained. A uniform prior was assigned to the intercept parameter $\beta_0$, while Gaussian (normal) priors were assigned to the regression coefficients. Also, the random effects were evaluated using zero-mean Gaussian functions with precisions $\tau_j$, which were assigned Gamma hyperprior distributions $(j \in \{B, \beta, l, e\})$. Then, with suitable prior and hyperprior parameters values, the predicted estimates of the bias-adjusted building intensity $\widehat{B}_i(i = 1, \ldots, n)$ were obtained by back-transforming the structured additive predictor $\eta_i^{(B)}$ defined in Eq. (3), that is,

$$\widehat{B}_i = g^{-1}\left(\eta_i^{(B)}\right) = \eta_i^{(B)} \qquad (3)$$

since $g(\bar{B}_i)$ is an identity link function.

### Step Two
As shown in Fig. 2, in step 2, the bias-adjusted settlement data $\widehat{B}_i$ serves as a model input for predicting population density and population counts across the entire set of spatial units of interest. Thus, following equation (S2, supplemental), the 'bias-corrected' population density $D_i$ which follows a Gamma probability distribution is estimated nominally as

$$D_i = \frac{C_i}{\widehat{B}_i}$$

where $C_i$ and $\widehat{B}_i > 0$ are the observed count of people, and the predicted (bias-adjusted) building intensity of the $i$ th spatial unit of interest, respectively. Note that while the methodology described here utilised population density $D_i$ defined as people per building, it could readily be applied to population density defined as people per unit settled area. The full hierarchical model set up is given below:

$$D_i \sim Gamma\left(\alpha_1^{(D)}, \alpha_2^{(D)}\right)$$

$$\bar{d}_i = \frac{\alpha_1^{(D)}}{\alpha_2^{(D)}}; \sigma_D^2 = \frac{\alpha_1^{(D)}}{\left(\alpha_2^{(D)}\right)^2}$$

$$g(\bar{d}_i) = \eta_i^{(D)}$$

$$\eta_i^{(D)} = \beta_0 + \sum_{k=1}^{K} \beta_k x_{ik} + \sum_{m=1}^{M} A_{im}\xi_m + \vartheta_{setTyp} + \vartheta_{Prov} + \vartheta_{setProv} + \vartheta_{source} + \zeta_i$$

$$\pi(\beta_0) \propto 1$$

$$\beta_k \sim Normal(\mu_\beta, 1/\tau_\beta)$$

$$\vartheta_l \sim Normal\left(0, \frac{1}{\tau_l}\right); l \in \{setTyp, Prov, setProv, source\}$$

$$\xi_m \sim SPDE$$

$$\zeta_i \sim Normal\left(0, \frac{1}{\tau_e}\right)$$

$$\tau_j \sim Gamma(\alpha_1^{(j)}, \alpha_2^{(j)}), where\ j \in \{B, \beta, l, e\} \qquad (4)$$

where $\alpha_1^{(D)}$ and $\alpha_2^{(D)}$ are the rate and shape parameters of the Gamma random variable $D_i$, respectively; $\bar{d}_i$ is the mean population density; the terms $\eta_i^{(D)}, \beta_0, \beta_1, \ldots, \beta_K, \tau, A, \xi_m$ and $\zeta_i$ are similarly defined as in Eq. (2); $\vartheta_l$ is a zero-mean Gaussian random effect term which in addition to accounting for the settlement type, Province and the interaction effects, also accounts for the effect of the differences in the data collection methods used by the urban listing and the malaria survey (*source*). Then, with appropriate priors (that is, priors that could not lead to model impropriety, *see 'Priors' Section* below), the predicted estimates of the population density are obtained by back-transforming the structured additive predictor $\eta_i^{(D)}$ defined in Eq. (2), that is,

$$\widehat{D}_i = \exp\left(\beta_0 + \sum_{k=1}^{K} \beta_k x_{ik} + \vartheta_{setTyp} + \vartheta_{Prov} + \vartheta_{setProv} + \vartheta_{source} \right.$$
$$\left. + \sum_{m=1}^{M} A_{im}\widetilde{\xi}_m + \zeta_i\right) \qquad (5)$$

Since $g(\bar{d}_i)$ is a log link function.

### Priors
The Bayesian statistical inference implementations in INLA[27,28] were based on the following appropriate priors and hyperprior values:

$$\beta_q \sim Normal(0, 0.01); q \in \{0, 1, \ldots, K\}$$

$$\vartheta_l \sim Normal(0, 1000); l \in \{setTyp, Prov, setProv, source\}$$

$$\tau_\varepsilon \sim Gamma(0.01, 0.01)$$

$$\tau_j \sim Gamma\left(1, 5 \times 10^{-5}\right); j \in \{B, \beta, l, e\}$$

Finally, the predicted population count $\widehat{C}_i$ is obtained as a product of the predicted population density and the bias-corrected settlement data, $\widehat{C}_i = \widehat{D}_i \times \widehat{B}_i$.

## Papua New Guinea simulation study

We carried out a simulation study to compare the performances of the BHM and TSBHM approaches. The simulation parameter values were chosen to obtain approximately the same total population count as in the real data model (Table S5–Supplemental Material). Thus, the underlying assumption in this simulation test is that the total population count is known. In order to more accurately mimic the sampling survey contexts, we allowed various proportions of the population to be missing at random (MAR). Then using our methodology, we examined how the 'true' total count initially simulated could be recovered over a range of survey coverage versus settlement data coverage combinations.

First, we used the GPS points of the centroids of the 32,100 census units across the 24 provinces in PNG to simulate the initial (full count) datasets (Fig. S6–Supplemental Materials). These include both the settlements (building) and population counts, and we also assumed that the settlements are perfectly observed at 100% level of observation. We then implemented various proportions of missingness across the survey coverage and satellite observation (proportion of observed settlements) coverages (see the simulation study section of the supplemental information document for more details).

## Papua New Guinea application

We applied our methodology to analyse the LLIN and USL survey datasets to produce modelled estimates of populations across PNG. The data were analysed using both the BHM and TSBHM methods. As noted earlier, Table S2 of the supplemental materials provides the list and description of the geospatial covariates that were eventually selected for the final models following initial GLM-based stepwise selection methods. In total, there were 13 covariates for the building intensity model and 15 covariates for the population density model. Then, both the BHM and TSBHM approaches described above were implemented using the PNG demographic and geospatial data, such that the predicted values of the bias-adjusted building intensity based on the best fit model were given by

$$\widehat{B}_i = \exp\left(\beta_0 + \sum_{k=1}^{13} \beta_k x_{ik} + \vartheta_{setProv} + \xi(s_i) + \zeta_i\right) \tag{6}$$

Also, the predicted population density based on the best fit model is given by

$$\widehat{D}_i = \exp\left(\beta_0 + \sum_{k=1}^{15} \beta_k x_{ik} + \vartheta_{setTyp} + \xi(s_i) + \zeta_i\right) \tag{7}$$

Finally, the predicted population count was given by $\widehat{C}_i = \widehat{B}_i \times \widehat{D}_i$.

## Model fit assessment and cross-validation

Model selection was based on the Widely Acceptable Bayesian Information Criterion (WAIC[38]) for goodness of fit, and the Conditional Predictive Ordinate (CPO[39]), which provides *leave-one-out* predictive measures for a single observation $y_i$ given other values $y_{-i}$ with $CPO_i = \pi(y_i|y_{-i})$. The model which provided the lowest WAIC values and the lowest negative sum of the natural logarithm of the CPO was retained as the best fit. In addition, we carried out in-sample and out-of-sample cross-validations to assess the predictive ability of the best of fit model. In the in-sample cross-validation, the model parameters were trained using all the available data points, after which data for a randomly selected 20% of the entire were selected for prediction. The out-of-sample cross-validation involved randomly dividing the entire dataset into a 20% test set and 80% training set (i.e., 80% of the data were used to train the model and the remaining 20% for prediction). In both in-sample and out-of-sample cross-validation we assessed model predictive ability using the model fit metrics outlined in Table S7 of the supplemental information document, which are the Mean Absolute

Error (MAE), Root Mean Square Error (RMSE), Absolute bias (Abias), and correlation coefficients between the observed and the predicted values.

Further checks were carried out to assess the performances of the BHM and TSBHM approaches in terms of reduction in relative error rates. Error rate (ER) was calculated as the difference between the observed total counts and predicted total counts divided by the observed total count:

$$ER^{(M)} = \frac{\widehat{y}_M - y_M}{y_M} \tag{8}$$

where *M* is a generic term from the set of methods {*BHM*, *TSBHM*}, so that $ER^{(BHM)}$ and $ER^{(TSBHM)}$ are the error rates corresponding to the BHM and TSBHM approaches, respectively; $\widehat{y}_{BHM}$ and $\widehat{y}_{TSBHM}$ are the predicted total counts from the BHM and TSBHM approaches, respectively; $y_{BHM}$ and $y_{TSBHM}$ are the corresponding observed total counts. Then, the relative error rate is $\theta^{(R)} = ER^{(TSBHM)}/ER^{(BHM)}$, so that a $\left(1 - \theta^{(R)}\right) \times 100\%$ reduction in relative error rate is calculated, with higher values representing better performances.

### Reporting summary

Further information on research design is available in the Nature Portfolio Reporting Summary linked to this article.

## Data availability

The input demographic and administrative datasets used in this study were provided by Papua New Guinea's National Statistical Office (NSO), while the input geospatial covariates were provided by WorldPop. The satellite-derived settlement dataset was provided by Planet (https://developers.planet.com/docs/data/planetscope/). Requests for these datasets should be directed to the appropriate owners. The small area scale population estimates produced for PNG are now published on the PNG NSO website here: https://www.nso.gov.pg/ and can easily be downloaded from WorldPop's data repository here: https://wopr.worldpop.org/?PNG/.

## Code availability

The R statistical programming code used in this study is available on the GitHub repository: https://github.com/wpgp/Small-area-population-estimation-from-health-intervention-campaign-surveys-and-partial-Observations[40].

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

## Acknowledgements

We acknowledge the financial support of the Australian Government and PNGAus partnership in supporting this work. The authors are also grateful to the PNG National Statistical Office and the UNFPA PNG, as well as the entire WorldPop team for their various contributions in making this study a success. We also acknowledge the support of Ramesh Nair at Planet on the training of the Planet settlement mapping model.

## Author contributions

C.C.N. prepared the first draft; C.C.N. and A.J.T. produced the second draft; C.C.N, A.B., J.J., O.Y., D.C., A.G., H.R.C., M.T., M.S., H.V., J.D., R.N., A.N.L., and. A.J.T. edited the manuscript; and J.J., M.T., J.D., R.N., and A.J.T. provided the input datasets; A.B., H.V., H.R.C. and T.A. processed the data; C.C.N. developed and implemented the model; A.N.L. and A.J.T. provided project oversight and A.J.T. acquired funding.

## Competing interests

The authors declare no competing interests.
