## [Transparent Peer Review file · Nature Communications]

Estimating small area population from health intervention campaign surveys and partially observed settlement data

Corresponding Author: Dr Chibuzor Nnanatu

Version 0:

Reviewer comments:

Reviewer #1

(Remarks to the Author)

Thank you for the opportunity to review this article. The authors propose a statistical model that takes satellite imagery, small-scale surveys, and geospatial covariates as inputs to produce a predicted building density and population density for areas with challenging terrain and/or areas affected by conflict. The utility of the proposed model is illustrated using data from Papua New Guinea.

Introduction

- The article is well-motivated, the analytic approach and the inherent challenges are well-justified. In the final paragraph of the introduction, it would be helpful if the authors clearly described what the output of their model will be with respect to Papua New Guinea. Currently, the text states that the analytic approach is applied to PNG without further description.

Data and Methods

- It would be helpful if the authors state whether their data are publicly accessible and/or derived from publicly accessible resources. This would help the reader gauge how reproducible their methods are on a different set of data.
- Please provide a brief (1-2 sentence) description of the "lowest available administrative units" for PNG. It is unlikely that readers will have a basic understanding of what this is. For ex, do these act more like US counties (geographic delineations with arbitrary population, relatively static in time), US census tracts (geographic delineations with approx. equal population, dynamic in time), etc. Figure 1 is helpful in this regard, but is hard to discern the boundaries of the admin units for much of the country.
- Although this reader appreciates the public health importance of malaria control, I am questioning whether the text around RAM contributes to the goals of the current article. If so, please describe how. If not, please consider revising or shortening this section.
- What was the goal/outcome of the manual comparison of satellite images and sample enumeration maps? Did the authors manually encode a geospatial variable or modify satellite images? Please make this more clear for reproducibility.
- Figure 2: this is a helpful figure. I am concerned about specific statistical elements of this figure that could either be corrected or removed/moved to the methods section. The notation: $w_1 = (\eta(B), \dots, f_{\text{spatial}}) \sim \text{GMRF}$ is confusing. The betas and f_{spatial} are all parts of the linear predictor η , although typically one specifies the GMRF only for the spatial component. The beta coefficients must not have follow a GMRF unless they are spatially-varying coefficients. Please reconsider this notation in Figure 2.
- The notation of random intercepts is non-standard for most literature. It appears the authors use $f()$ similar to what is used in the INLA formula. Please consider standardizing this notation to something more recognizable to future readers.
- Figure 2: The Poisson distribution, conditional on the spatial effect, in Step 2 seems appropriate. Please comment on whether the log-Normal distribution for building intensity is restrictive, perhaps in supplementary methods. For example, why not a Skew-Normal or a Gamma? Is there evidence that the log-Normal would work well for other regions of the world besides PNG?
- SPDE Mesh Construction: please provide enough details in supplementary methods such that the INLA SPDE Mesh construction is reproducible. As the authors are aware, the mesh has a non-trivial impact on the modeling results.

Simulation Results & Discussion of Simulation

- The MAR as opposed to MCAR assumption necessary to justify this model is important and is well-described.
- Figure 3 makes a compelling case for the two-stage model proposed by the authors.
- Consider moving Figure 4 to the supplement; I'm not sure it contributes significantly to the article.

* PNG Application *

- Table 1: I follow all model accuracy metrics except the correlation coefficient. It makes sense it would be near 1 for in-sample model comparisons, which calls into question the utility of providing these columns if the goal of modeling is to make an out-of-sample prediction. Likewise, why is the correlation coefficient lower for the two-stage than the one-stage model with respect to density?
- Figure 6: I take it that provinces are larger than the geographic units used, but it bears repeating. Consider using a different color for the credible intervals - black on blue is hard to see.

* Discussion *

- Under the strengths of the analytic approach, I feel that the lack of discussion around the potential use of posterior distributions for better selection of target for intervention (e.g., via exceedance probability) is a missed opportunity. Likewise, INLA is a rather mature software package at this point; thus, its outputs can be standardized to fit into reporting and tracking systems used by various governments, WHO, UNESCO, etc. I would add these aspects to the strength of the modeling approach.
- Please add a paragraph as to the anticipated challenges of applying the analytic technique to a different location. For example, are there additional challenges to mapping small island nations in the Pacific? Mountainous areas in Central Asia? Parts of Siberia that may be covered in snow nearly all year? I would like to see additional text on how generalizable the methods are to other areas of the world.

(Remarks on code availability)

The code is appropriate for the article, is well-documented, and appears reproducible.

Reviewer #2

(Remarks to the Author)

See attached PDF with comments for authors.

(Remarks on code availability)

Version 1:

Reviewer comments:

Reviewer #1

(Remarks to the Author)

I have reviewed the changes made to the manuscript and believe my comments have been addressed. Thank you for the opportunity to review this important work.

(Remarks on code availability)

Code previously reviewed and found to be adequate for reproducibility.

Reviewer #2

(Remarks to the Author)

I thank the authors for their detailed responses to my earlier comments. I have just a few more comments, below:

1. p. 9, Fig. 2. Thank you for updating this in response to my previous comment. However, in Step 2, should " B_i " be " \hat{B}_i "? I.e., the estimates from Step 1 plugged in? If not, then it reads as if B_i is a random variable with the distribution in Step 1. There, you have $\log(B_i) \sim \log\text{-Normal}(b_i, \text{sig}^2_b)$. If $D_i \sim \text{Gamma}(a_1, a_2)$ and C_i is defined as the product of D_i and B_i and given a Poisson distribution, you'd need to show that the product of a gamma and log-Normal is, indeed, a Poisson, which I'm not sure it is.

2. p.9, Fig. 2 (again) + Suppl. Mat., p. 7. B_i is modelled as a continuous variable in Fig. 2 (log-Normal), but is described as "the number of buildings" (i.e., a count) in the supplemental material. Table S1c describes "B" as both an intensity and a count, but these are different quantities (one is continuous, the other is discrete). I don't think it's a problem to use a continuous distribution when the counts are large, but you use a discrete distribution for population counts, so why the

difference? Furthermore, in the manuscript, p. 10, I think you are saying that B_i ranges from 0 to 88,237. Are these counts of buildings? I think I'm just confused by your use of the word "intensity" for B_i . In the supplemental (p. 7) you say " B_i is the number of buildings". You call the C_i (the number of people) "counts" so, why not for B_i as well? It might also help to put in the main article that that D_i is a density in the sense of people-per-building, as opposed to something else like people per unit of area, for example.

3. Suppl. Mat., Table S1c, p. 7. Thanks for adding this table, it's very useful. However, Bhat, Chat, Dhat are all described as "Data", but if they're predicted values from a model they can't be observed, so "Data" doesn't seem right. Can this be corrected?

(Remarks on code availability)

Version 2:

Reviewer comments:

Reviewer #2

(Remarks to the Author)

Dear authors,

Thanks for addressing my final comments and for the opportunity to review. I have no further remarks.

(Remarks on code availability)

REVIEWERS COMMENTS

Reviewer #1 (Remarks to the Author):

Thank you for the opportunity to review this article. The authors propose a statistical model that takes satellite imagery, small-scale surveys, and geospatial covariates as inputs to produce a predicted building density and population density for areas with challenging terrain and/or areas affected by conflict. The utility of the proposed model is illustrated using data from Papua New Guinea.

RESPONSE: Thank you.

Introduction

- The article is well-motivated, the analytic approach and the inherent challenges are well-justified. In the final paragraph of the introduction, it would be helpful if the authors clearly described what the output of their model will be with respect to Papua New Guinea. Currently, the text states that the analytic approach is applied to PNG without further description.

RESPONSE: Thank you. We have added some texts within the final paragraph of the introduction section to describe the model outputs that are specific to PNG context. These are highlighted in yellow in the Revised Manuscript – “...by producing model-based small area population estimates for Papua New Guinea..”

Data and Methods

- It would be helpful if the authors state whether their data are publicly accessible and/or derived from publicly accessible resources. This would help the reader gauge how reproducible their methods are on a different set of data.

RESPONSE: Thank you. The input demographic and administrative datasets used in this study were provided by Papua New Guinea's National Statistical Office (NSO; <https://www.nso.gov.pg/statistics>), while the input geospatial covariates were provided by WorldPop (www.worldpop.org). The satellite-derived settlement dataset was provided by Planet (<https://developers.planet.com/docs/data/planetscope/>). Requests for these datasets should be directed to the appropriate owners. However, to facilitate the review processes and promote knowledge sharing, we have deposited the scripts for R codes and the key input datasets used for the study on the GitHub repository where it is also easily accessible: <https://github.com/wpgp/Small-area-population-estimation-from-health-intervention-campaign-surveys-and-partial-Observations/>. This is now stated in the data and code availability sections at the end of the manuscript.

- Please provide a brief (1-2 sentence) description of the "lowest available administrative units" for PNG. It is unlikely that readers will have a basic

understanding of what this is. For ex, do these act more like US counties (geographic delineations with arbitrary population, relatively static in time), US census tracts (geographic delineations with approx. equal population, dynamic in time), etc. Figure 1 is helpful in this regard, but is hard to discern the boundaries of the admin units for much of the country.

RESPONSE: Thank you. The lowest administrative units in Papua New Guinea at which the population data are available is the 'Census Unit'. The model input data are prepared at the census unit levels. This was previously missed in the caption of Figure 1, but we have now revised the caption of Figure 1 to include 'Census Unit' which is also highlighted in Yellow in the Revised Manuscript, to help in improving the understanding of the readers.

- Although this reader appreciates the public health importance of malaria control, I am questioning whether the text around RAM contributes to the goals of the current article. If so, please describe how. If not, please consider revising or shortening this section.

RESPONSE: Thank you. We totally appreciate this comment by the reviewer, and we have now revised (shortened) the paragraph focusing on the RAM data description and highlighted in yellow in the Revised Manuscript – “...*Rotarians Against Malaria (RAM) is a PNG based Non-Governmental Organisation (NGO) that works in collaboration with the National Department of Health (NDOH) along with other organisations such as UNICEF, WHO and church groups to reduce and control the impact of malaria in PNG. RAM's primary activity is the distribution of Long-Lasting Insecticidal Nets (LLINs) across PNG with each village visited once every three years. Population surveys are carried out routinely prior to net distribution to ensure every household gets adequate number of nets. Sometimes, it is not possible to visit all the villages in each round of survey due to factors such as villages that are too remote, shortage of funds, and other factors...*”

- What was the goal/outcome of the manual comparison of satellite images and sample enumeration maps? Did the authors manually encode a geospatial variable or modify satellite images? Please make this more clear for reproducibility.

RESPONSE: Thank you. The goal was to highlight how settlements were obscured by tree canopies, and we have added clarification to the main text on this. The sample enumeration data provided 'ground-truth' data evidence that made it possible to compare the number of settlement structures observed through the satellite imagery at a given spatial location with the actual number of people within the enumeration data for that location. The assumption is that higher number of settlements is associated with more people (or higher population density) and vice versa. Thus, in remote areas (e.g., Figure S1) where there are several tree canopies, fewer number of settlements in comparison to the number of people within the

'ground-truth' enumeration data suggests partial or incomplete satellite observations of the settlements.

The comparisons were done by simply visualising the distribution of the enumeration data at a specific location overlaid with the satellite imagery of settlements within the same location. Given the known dense vegetation cover of large parts of Papua New Guinea, we hypothesised that some settlements may not have been captured in satellite-based building detections (such as the Planet data). This was explored through manual comparisons of the Planet data and recent high resolution satellite images against the sample population enumeration data (see Figure S1 of the supplementary materials for an example). These comparisons highlighted forested rural areas where substantially larger numbers of people were enumerated compared to area of settlement and/or buildings mapped, confirming that settlements had likely been missed. Additional comparisons against other satellite-derived settlement mapping datasets, e.g., the Global Human Settlement Layer (GHSL2123), found similar or worse issues in terms of missing settlement

- Figure 2: this is a helpful figure. I am concerned about specific statistical elements of this figure that could either be corrected or removed/moved to the methods section. The notation: $w_1 = (\eta(B), \dots, f_{\text{spatial}}) \sim \text{GMRF}$ is confusing. The betas and f_{spatial} are all parts of the linear predictor η , although typically one specifies the GMRF only for the spatial component. The beta coefficients must not have follow a GMRF unless they are spatially-varying coefficients. Please reconsider this notation in Figure 2.

RESPONSE: Thank you. We appreciate the reviewer for this important observation. Although, within the context of INLA modelling framework, all the unknown parameters are classified as latent Gaussian Markov Random Fields (GMRF) implied through the sparse precision matrix Q – the inverse of the dense covariance matrix Σ (e.g., Rue & Held, 2005; Rue et al. 2009), that is, $w_1 \sim \text{Normal}(0, Q(\theta))$. However, the reviewer was right to say that it was confusing going by the way it was linked to the mesh by the last sentence within the caption of Figure 2. The GMRF within Figure 2 has now been removed and the caption has also been revised – “... C_i is the count of people in census unit i ; D_i and B_i are the population density and building intensity for census unit i , respectively; $\vartheta_{(\cdot)}$ are zero-mean Gaussian random intercepts for settlement types ($setTyp$) and settlement type versus Province interactions ($setProv$). The spatial random effect ξ is modelled via Gaussian Markov random field (GMRF).”

- The notation of random intercepts is non-standard for most literature. It appears the authors use $f(\cdot)$ similar to what is used in the INLA formula. Please consider standardizing this notation to something more recognizable to future readers.

RESPONSE: Thank you. We have revised the notations throughout the 'Revised Manuscript' and the $f()$ notation are now changed with $\vartheta_{()}$ for each random effect of interest.

- Figure 2: The Poisson distribution, conditional on the spatial effect, in Step 2 seems appropriate. Please comment on whether the log-Normal distribution for building intensity is restrictive, perhaps in supplementary methods. For example, why not a Skew-Normal or a Gamma? Is there evidence that the log-Normal would work well for other regions of the world besides PNG?

RESPONSE: Thank you. The log-normal distribution for the building intensity is non-restrictive, and any other positively skewed distribution (including skewed-normal, log-logistic, and Gamma densities) could be used. However, from preliminary analyses, the use of log-normal distribution appears to work best for our present context. We have updated the methods section of the 'Revised Manuscript' under Step-One (highlighted in yellow) to add these important texts – "...*Note that apart from the normal distribution, any other skewed distribution (including skewed-normal, log-logistic, and Gamma densities) could be used to model the logarithm of the building intensity. However, the use of the normal distribution appears to work best in our context*"

- SPDE Mesh Construction: please provide enough details in supplementary methods such that the INLA SPDE Mesh construction is reproducible. As the authors are aware, the mesh has a non-trivial impact on the modeling results.

RESPONSE: Thank you. All the R codes used in the manuscript which allows for reproducibility are freely available on the GitHub repository: <https://github.com/wpgp/Small-area-population-estimation-from-health-intervention-campaign-surveys-and-partial-Observations/>. We have also added more text under Figure S3 to specify some key parameters of the inla.mesh functions used in constructing the mesh.

Simulation Results & Discussion of Simulation

- The MAR as opposed to MCAR assumption necessary to justify this model is important and is well-described.

RESPONSE: Thank you. We appreciate the reviewer for this kind comment.

- Figure 3 makes a compelling case for the two-stage model proposed by the authors.

RESPONSE: Thank you.

- Consider moving Figure 4 to the supplement; I'm not sure it contributes significantly to the article.

RESPONSE: Thank you. We believe that Figure 4 provides an important summary of the reduction in relative error rates attributable to the 'new' TSBHM method over the 'old' BHM method across various scenarios. This is a key result in the study, and we strongly believe that Figure 4 serves to deepen understanding and appreciation of the work for non-technical readers.

* PNG Application *

- Table 1: I follow all model accuracy metrics except the correlation coefficient. It makes sense it would be near 1 for in-sample model comparisons, which calls into question the utility of providing these columns if the goal of modeling is to make an out-of-sample prediction. Likewise, why is the correlation coefficient lower for the two-stage than the one-stage model with respect to density?

RESPONSE: Thank you. We totally agree with the reviewer that the in-sample accuracy metrics are expected to be very good. However, it is possible that this is not usually the case especially for poorly performing models. The use of the in-sample accuracy metrics helped us to compare the model accuracy metrics for the out-of-sample cross-validation, since out-of-sample metrics that appear much worse than the in-sample metrics values are indicative of a model with poor predictive ability. For example, in Table 1, it is easy to see that for both the BHM and TSBHM methods, the out-of-sample cross validation metrics were not worse than the in-sample metrics, thus, suggesting that our proposed TSBHM approach has a significantly high predictive ability. Additionally, the overall best fits were provided by the TSBHM when compared to the BHM metrics.

The lower correlation coefficient of the TSBHM (0.792) as against the 0.992 of the BHM model for the density models could be due to some random variations within the cross-validation samples. However, when compared across the values of the MAE, RMSE and absolute bias (Abias), the TSBHM provided the best fit against the BHM model throughout.

Given the reviewer's questions, we have endeavoured to clarify all of this in the texts of the revised manuscript to help other readers.

- Figure 6: I take it that provinces are larger than the geographic units used, but it bears repeating. Consider using a different color for the credible intervals - black on blue is hard to see.

RESPONSE: Thank you. We have now reproduced and replaced Figure 6 to make the 95% credible interval lines more visible.

* Discussion *

- Under the strengths of the analytic approach, I feel that the lack of discussion around the potential use of posterior distributions for better selection of target for intervention (e.g., via exceedance probability) is a missed opportunity. Likewise, INLA is a rather mature software package at this point; thus, its outputs can be standardized to fit into reporting and tracking systems used by various governments, WHO, UNESCO, etc. I would add these aspects to the strength of the modeling approach.

RESPONSE: Thanks for this important point. We have now revised the discussion section of the 'Revised Manuscript' to include texts around the potential of the posterior samples, including in targeted interventions – *“Moreover, the model outputs (including the posterior distributions) can serve to support governance, reporting and tracking systems used by governments and international agencies, and are already in use for country planning in PNG (<https://www.nso.gov.pg/statistics/population/>).*

- Please add a paragraph as to the anticipated challenges of applying the analytic technique to a different location. For example, are there additional challenges to mapping small island nations in the Pacific? Mountainous areas in Central Asia? Parts of Siberia that may be covered in snow nearly all year? I would like to see additional text on how generalizable the methods are to other areas of the world.

RESPONSE: Many thanks for this very crucial point. We have revised the relevant sections of the 'Revised Manuscript' highlighted in yellow to include the anticipated challenges/limitations of our techniques and how they could be adapted for other contexts – *“we anticipate that challenges will remain in mapping populations in certain types of terrains, such as mountainous, desert and snow-covered areas, where the accuracy of satellite-based human settlement mapping tends to be lower due to topographic variations and the similarities between human settlements and the surrounding landscapes.”*

Reviewer #1 (Remarks on code availability):

The code is appropriate for the article, is well-documented, and appears reproducible.

RESPONSE: Thank you. We appreciate the reviewer for this kind comment. Yes, the codes are easily accessible, and the results are reproducible.

Reviewer #2 (Remarks to the Author):

Major Points

1. BHM & TSBHM:

1.1 I found it hard to keep track of what quantities are directly observed (i.e., the “data”) and which are unobserved (i.e., model parameters). E.g., I think the B_i in eqn (1) are the building intensity observations from Planet, while \bar{B}_i is a model parameter (unobserved). Similarly, I assume that C_i are the observed population counts from the survey data (but then what are Y_i ?). It might be clearer to use, say, roman letter for data and greek letters for unobserved model parameters. Or capital vs lower case, or something else systematic. A summary table of symbols explaining what each is and whether it is directly observed or not, fixed or random, would be helpful.

RESPONSE: Thanks a lot for these important observations. We have now revised the notations accordingly using \bar{b} and \bar{d} for the mean parameters of building intensity (B) and population density (D), respectively. We have updated the equations throughout in the ‘Revised Manuscript’ including Figure 2 along with the typographical errors in the caption. These changes and others are highlighted in yellow within the ‘Revised Manuscript’. Additionally, we have created a table of symbol in **Table S1c** of the supplementary document to provide a standardized system of data versus parameters notation used in the manuscript.

1.2 You list two sources of population estimates from surveys: urban structural listings and malaria surveys. How did the two different sets of counts enter the model? Is there a model parameter for survey type to , for example, account for any difference in accuracy by survey type?

RESPONSE: Thank you for this important point. We conducted series of preliminary analyses to assess the influence of the differences in data collection strategies utilised by the different data sources. This was done by including a data source random effect (*source*) to the BHM model. However, model fit checks indicated that the model without the data source random effect provided a better fit with smaller DIC values (- 217,407.9 – without data source random effect, and -216,071.1 – with data source random effect). Thus, the data source random effect was dropped for the TSBHM model to enable direct comparison between the BHM best fit model and the corresponding TSBHM-based model. This important information was previously missed in the manuscript, but we have now added texts on this within the ‘Papua New Guinea application’ results section of the ‘Revised Manuscript’

1.3 How does the building intensity model (Step 1) actually correct the bias due to tree cover? It seems this must always be negative as tree cover obscures some structures so they are undercounted. Is tree cover a covariate in the model? Shouldn’t only those cells that are under trees should be corrected? As written, it looks like the bias-corrected building intensities will be averages

over intensities from both tree covered and non-tree covered mesh nodes, within observed combinations of covariates. This would under-correct tree covered location but over-correct uncovered locations.

RESPONSE: Many thanks for these important questions. Yes, the building intensity model was able to correct biases in the settlement by exploiting the relationship between the geospatial covariates used in the model to predict building intensity values at locations under tree canopy cover as well as other locations with partly obscured observations. Following from equation S2 of the supplementary document, the variance of the population density increases exponentially with unrealistically small values of the building intensity, which will in turn lead to biased and highly inflated population counts. By subjecting the imperfectly observed building intensity to statistical modelling, we were able to predict the missed building intensities which then directly fed into Step 2 leading to much improved estimates of population.

Although a tree cover measure was not used as a covariate in the statistical model, other closely related/proxy covariates (e.g., distance to aquatic vegetation areas, distance to cultivated areas, distance to shrub area edges, etc) listed in Table S1 of the supplemental document were used. In addition, the bias-correction was applied across all locations including observed and partially observed locations. The log transformed values of the observed building intensity were modelled using Gaussian density and all the back-transformed (exponentiated) values of the building intensities returned positive values. Another advantage of the Step 1 model is that it enabled us to predict building intensities across completely obscured locations thereby enabling the estimation of population counts within such areas.

We agree with the reviewer that there are chances that the building intensity model could under-correct tree covered locations but over-correct uncovered locations, however, by predicting the building intensities while drawing upon their relationships with a suite of geospatial covariates and spatial autocorrelation, such chances are most likely to be very low with insignificant impacts.

Given the reviewer's questions, we have endeavoured to clarify all of this in the text to help other readers.

1.4 How is the uncertainty about B_i from Step 1 propagated through to Step 2?

As written, it looks like the point estimates (without uncertainty) from step 1 are just plugged in to step 2.

RESPONSE: Thank you. Yes, within the current study, we have only utilised the posterior mean of the predicted building intensity obtained in Step 1 to estimate population density in Step 2. We totally agree with the reviewer that integrating the uncertainty in the Step 2 would be interesting and we have now acknowledged this

as a limitation in the study within the Discussion section of the 'Revised Manuscript' – *“...within the current study, we have only utilised the posterior mean of the predicted building intensity obtained in Step 1 to estimate population density in Step 2. However, it would be interesting for future studies to integrate the Step 1's estimates of uncertainties of the building intensity in the estimation of population density in Step 2 for potentially more robust estimates”*.

1.5 Step 2: “with appropriate priors (that is, priors that will not lead to model impropriety)”. These priors need to be described somewhere.

RESPONSE: Thank you for this crucial point. We have now updated the 'Revised Manuscript' within 'Step two' section to include the actual priors that were used for the Bayesian inference implementation within the INLA framework.

2 Geospatial covariates:

2.1 If Table S1 has the “finale model covariates”, what was the initial candidate set?

RESPONSE: Thank you. The initial candidate set of the covariates used in the preliminary models (both building intensity and population density) is presented in Table S1b of the supplementary document (under Table S1).

2.2 Was the stepwise selection done using the 'raw' data (i.e., the data subject to bias due to tree cover, etc.)? For the final model, was the procedure the same as described in the simulation study?

RESPONSE: Thank you. The stepwise selection was done separately for both the building intensity and the population density models based on the 'raw' data. Although an alternative approach would be to use the 'raw' data for the BHM and the 'bias-corrected' data for the TSBHM, however, this would make it difficult to carry out a direct comparison of the performances of the BHM versus the TSBHM approaches.

After the covariate selection, the main distinction between the BHM and TSBHM approaches stemmed from the response variables (population density) used for the implementation of the Bayesian hierarchical models in Step 2. Specifically, the population density variable for the BHM was defined with the raw building intensity data, while the population density for the TSBHM model was defined in terms of the bias-corrected building intensity data.

The BHM-based nested models were first tested after incorporating all key random effects (e.g., data source, settlement type, province, settlement type versus province interactions), and spatially varying and spatially independent random effects. The

aim of this second level of tests is to see which of the random effects combinations best described the relationship between the population density and the set of 'best' geospatial covariates.

Following model fit checks (primarily using DIC values), the best BHM-based model is then replicated using the TSBHM approach. Both models are then checked further in terms of accuracy and predictive ability using a constellation of model fit assessment metrics (e.g., Absolute bias, MAE, RMSE, reduction in relative error rates, etc).

2.3 Were the same covariates used for the one-step and two-step models? This could be problematic if a particular covariate pattern implies contradictory outcomes in the two steps. Are the Beta coefficients at least different between the two steps?

RESPONSE: Thank you. Initially, the same set of covariates were used for the stepwise selection but both models ended up with different sets of 'best' covariates for the building intensity model and the population density model although with some covariates being retained for both models (please see, Table S1 of the supplementary document). This makes sense, because while we expect slightly different relationships between the set of covariates and the building intensity/population density, it is also expected that some covariates will be strongly correlated with (strong predictors of) both building intensity and population density. For example, distance to local roads and distance to aquatic vegetation areas were found to be key predictors of both building intensity and population density. Whereas, distance to health providers significantly predicted population density and not building intensity; while distance to cultivated areas was a significant predictor of building intensity but not population density (Table S1).

Also, yes, the Beta coefficients (for covariates appearing in the two models) are different, as expected, since they were based on different response variables.

3. Simulation Study (Appendix):

3.1 The point of the simulation was not clear until the end of the paper, especially the intent to compare the BHM and TSBHM results. You could make this clearer earlier on.

RESPONSE: Thank you. We have included this important point within the 'Revised Manuscript' highlighted in yellow – *"We carried out an extensive simulation study to compare the performances of the BHM and TSBHM approaches"*

3.2 You state “simulation parameters values were chosen to obtain approximately the same total population count as in the real data model”, which I assume means the estimated population from the full data set.

RESPONSE: Thank you. Yes, that's correct. The statement was referring to the estimated population count but it was for the full observation at 100% coverage with perfect observation of human settlements (i.e., when the entire population were assumed to be completely observed with no missingness). We have clarified this in the main texts of the 'Revised Manuscript'.

But then you multiply the survey and the building counts by 20, 30, per cent. Did you then expect to recover the original total population counts?

RESPONSE: Thank you. Yes, that's correct. The full population simulated at 100% coverage with perfect observation of human settlement was tweaked such that only a fraction ($p\%$) of the population and a fraction ($b\%$) of the human settlement (or building intensity) were observed. Each fraction of the population was obtained by multiplying the full population count by p (for a $p\%$ survey coverage) and each fraction of the building intensity was obtained by multiplying the full building intensity by b (for a $b\%$ building intensity observation), where $p \in \{0.2, 0.4, 0.6, 0.8\}$ and $b \in \{0.65, 0.7, 0.75, 0.8, 0.85, 0.9, 0.95\}$. The idea was to see how each of the BHM and TSBHM was able to recover the 'true' total population count (at 100% full observation under perfect settlement data observation) over the different combinations of missingness. The closer the estimated total population is to the 'true' total population the better performing is the approach. We have clarified this in the 'Revised Manuscript'.

The sentence “the 'true' population is taken to as 11.643,074 people and model performances are adjudged by high close the total population estimates are to the 'true' value across the various proportions of missingness” suggests the former. Put another way, are the population totals in Table S3 inputs to the simulation study or model outputs?

RESPONSE: Thank you. Yes, the population totals in Table S3 are the totals of the simulation counts used as the model inputs. We have clarified this in the 'Revised Manuscript'.

3.3 Step 2: The spatial covariates, x_1, \dots, x_5 are resampled at each iteration. There are twenty of these in Table S1 but the x_k sequence only goes up to 5. What about the rest?

RESPONSE: Thank you. For ease of exposition, only 5 geospatial covariates were simulated and used for the purpose of the simulation study. Thus, these covariates were randomly generated and are different from the real data implementation covariates presented in Table S1. We have clarified this in the ‘Revised Manuscript’.

Moreover, these look like they’re probably correlated from their descriptions (e.g., distance to local roads and distance to main roads). If you just simulate these independently, how can you be sure you end up with combinations that make physical sense?

RESPONSE: Thank you. Yes, some of the covariates used in the real-data applications show strong correlations but the use of the stepwise selection approach in conjunction with the vif (variance inflation factor) checks ensured that the potential issues of multicollinearity were reduced to the barest minimum.

Although the 5 geospatial covariates used in the simulation study for learning purposes were simulated independently of each other (mainly to avoid multicollinearity issues), the values of the corresponding simulated building intensity and population counts were dependent on their correlations with the geospatial covariates. We have clarified this in the main text.

3.4. Step 7: What does “posterior simulation of the results” entail?

RESPONSE: Thank you. The posterior simulation provides a sampling distribution of the posterior samples to further improve the accuracy of the posterior estimates obtained from the models, and to obtain uncertainties around aggregated total counts. This is because while it is straightforward to obtain the overall total counts from the aggregated posterior means, aggregation of the posterior lower and upper bounds of the 95% credible intervals in order to obtain measures of uncertainty around the aggregated totals is technically meaningless. To do this, we require multiple samples of the posterior estimates so that we can have an aggregated total from each of the samples. Then we can calculate the mean and the 95% credible intervals of the sample distribution of the aggregated totals. We have clarified this in the main text.

3.5 How are the coverages p and b applied in the simulation? These symbols don’t appear anywhere in steps 1 - -7.

RESPONSE: Thank you. As already mentioned above, p is the proportion of the total population observed with $p = 1$ indicating a full count or 100% observation. Also, b is the proportion of the human settlement (building intensity) observed with $b = 1$ indicating a perfect observation (no missingness or no obscurity of the satellite

observation). The values are taken from Table S2. We have highlighted in yellow within step 1 and step 8 of the 'Simulation Study Steps' of the supplementary document where the use of p and b were mentioned.

3.6 What are the plotted points in Figure S6? What do the fitted lines represent?

RESPONSE: Thank you. The plotted points in Figure S6 are the simulated (observed) versus predicted population counts obtained across the various proportions of the survey coverage $p \in \{0.2, 0.4, 0.6, 0.8, 1\}$ when the settlement data were completely observed ($b = 1$). The points are presented along with the 95% credible intervals as point ranges. The fitted lines are the regression lines (mean) for each dataset simulated for each of the p values.

We have added 'simulated (observed)' to the caption of Figure S6 of the supplementary document to avoid confusion.

4. Papua New Guinea Application:

4.1 Figure 6: What's the explanation for the systematically higher estimates from the BHM model?

RESPONSE: Thank you. This crucial observation underscores the importance of developing the TSBHM approach where we had to correct for biases within the building intensity first before the estimation of population counts in Step 2. The reasoning can be seen from equation (S2), where the variance of the population density increases exponentially over unrealistically small values of the building intensity, which in turn leads to inflated (biased) population counts. When we correct for these potential biases in the settlement data, we gain a large accuracy advantage by being able to predict values of the settlement data where they were only partially observed. We have clarified this in the 'Revised Manuscript'.

Is it that the BHM underestimates B_i because buildings under tree cover are not counted?

RESPONSE: Thank you. Yes, this is correct and also in line with the explanations above.

4.2 How do you estimates of the total PNG population compare with the official estimates from the NSO and international organizations?

RESPONSE: Thank you. The real data applications to the PNG context were implemented in collaboration with the PNG's National Statistical Office (NSO) technical team. Thus, the population estimates produced in this study are the same as those published within the PNG's NSO website

<https://www.nso.gov.pg/statistics/population/>. The data are also available on WorldPop's data repository here <https://wopr.worldpop.org/?PNG/Population>. Our national total estimate using this modelling approach is 11.78 million people (95% intervals: 11.64M-12.03M), while the latest estimates for the same year from the UN World Population Prospects 2024 edition (<https://desapublications.un.org/publications/world-population-prospects-2024-summary-results>) are 10.84 million and the US Census Bureau estimates 9.59M million for 2022. While our principal focus here was on subnational distributions, this shows that our national estimates were in line with those from other sources. Moreover, by using recently observed input datasets, our methodology was able to capture recent changes in PNG's unique population structure and distribution at very small area units. Such spatially detailed fine-grained population data required for several health intervention campaigns are mostly lacking in the other sources. Additionally, unlike most other sources which only provided point estimates, the Bayesian statistical modelling approach we utilised ensured that we have estimates of uncertainties which are vital for planning purposes.

Other Points

1. p.10: In Figure 2, Step 2, you have $Y_i \sim \text{Poisson}(D_i * B_i)$, then below you have $D_i = Y_i / B_i$. This would imply that $Y_i \sim \text{Poisson}(Y_i)$ which doesn't make sense. Is something wrong here? In the caption you also have $Y_i = C_i$. Why the two symbols? If one is 'data' and the other is a model parameter it might be useful to make this clearer.

RESPONSE: Thank you. We have now revised these within the 'Revised Document' and the changes are highlighted in yellow throughout. Changes were also made within the supplementary document.

2. p.11: 1st para: You have "the resampled Planet settlement data was resampled". How many times was it resampled?

RESPONSE: Thank you. This was a typo which is now corrected in the 'Revised Manuscript' and highlighted in yellow.

3. p.11: 1st para: "The BHM scheme assumes that the input settlement building intensity B_i is perfectly observed". I assume this is the single-stage model and that you're comparing it to the two-stage model. Maybe make this a bit clearer since both your models are technically BHMs, e.g., add a citation reference to the "old" BHM you're improving.

RESPONSE: Thank you. Yes, this is correct. We have added the 'old' BHM citations at the appropriate point within the 'Revised Manuscript' as kindly suggested.

REFERENCES

- Rue, H., Held, L. (2005). Gaussian Markov random fields. *Theory and applications*. Chapman & Hall.
- Rue, H., Martino, S., Chopin, N. (2009). "Approximate Bayesian Inference for Latent Gaussian Models by Using Integrated Nested Laplace Approximations." *Journal of the Royal Statistical Society, Series B* 71 (2): 319–92

REVIEWERS COMMENTS

Reviewer #1 (Remarks to the Author):

I have reviewed the changes made to the manuscript and believe my comments have been addressed. Thank you for the opportunity to review this important work.

Reviewer #1 (Remarks on code availability):

Code previously reviewed and found to be adequate for reproducibility.

Reviewer #2 (Remarks to the Author):

I thank the authors for their detailed responses to my earlier comments. I have just a few more comments, below:

1. p. 9, Fig. 2. Thank you for updating this in response to my previous comment. However, in Step 2, should " B_i " be " \hat{B}_i "? I.e., the estimates from Step 1 plugged in? If not, then it reads as if B_i is a random variable with the distribution in Step 1. There, you have $\log(B_i) \sim \log\text{-Normal}(b_i, \sigma^2_b)$. If $D_i \sim \text{Gamma}(a_1, a_2)$ and C_i is defined as the product of D_i and B_i and given a Poisson distribution, you'd need to show that the product of a gamma and log-Normal is, indeed, a Poisson, which I'm not sure it is.

RESPONSE: Thank you very much for flagging this important omission. Yes, we agree that it is important to clearly differentiate between the notations for the estimated values and the random variable itself. We have now updated Figure 2 and wherever it was omitted in the manuscript as well as in the supplementary document. Specifically, in Figure 2, the estimated values of the settlement data are now denoted by \hat{B}_i in Step 2, while B_i denote the corresponding random variable in Step 1. All the changes are highlighted in yellow wherever it applies in both the revised manuscript and the supplementary document.

2. p.9, Fig. 2 (again) + Suppl. Mat., p. 7. B_i is modelled as a continuous variable in Fig. 2 (log-Normal) but is described as "the number of buildings" (i.e., a count) in the supplemental material. Table S1c describes "B" as both an intensity and a count, but these are different quantities (one is continuous, the other is discrete). I don't think it's a

problem to use a continuous distribution when the counts are large, but you use a discrete distribution for population counts, so why the difference?

RESPONSE: Thank you very much for highlighting the need for this important clarification. Recently, building counts derived from satellite-based building footprint maps using Machine Learning algorithms have become the main input human settlement data in 'bottom-up' population models¹⁻³. In the supplementary material, we provided the generic background framework where building count was the main input settlement data, thus, the use of 'building count' within the supplementary document. However, in the revised manuscript, we have used building intensities (range: 0 – 88,237) derived from the Planet satellite data (because these were the most recent and reliable data on building available in PNG at the time of writing, as outlined in the manuscript) as the proxy measures of the building counts, such that higher values indicated higher likelihood of building presence (please see the 'Settlement maps' section of the revised manuscript – pp. 7-8). While there is need to ensure clarity in terms of the use of terms, we also believe that it is important to make the distinctions that the building intensities were not the same as building count but that both are used as indicators of human settlements. We have now added the below texts to the manuscript in the 'Methods' section under "**Two-step Bayesian hierarchical modelling (TSBHM) framework for bottom-up population modelling with partially observed settlement data**" to clarify this important point:

"Please note that building intensity and building counts, although not the same, have been used interchangeably in the manuscript and in the supplementary document to serve as proxy to human settlements within the population models."

Furthermore, in the manuscript, p. 10, I think you are saying that B_i ranges from 0 to 88,237. Are these counts of buildings? I think I'm just confused by your use of the word "intensity" for B_i . In the supplemental (p. 7) you say " B_i is the number of buildings". You call the C_i (the number of people) "counts" so, why not for B_i as well? It might also help to put in the main article that that D_i is a density in the sense of people-per-building, as opposed to something else like people per unit of area, for example.

RESPONSE: Thank you very much. We agree with the Reviewer that the values of both the building intensity and the building count are discrete counts which could be modelled with Poisson distribution as was done for the population counts. However, preliminary exploratory checks indicated that the log transformed version of the building count/building intensity data was continuous, normally distributed and provided good fit. Then using the identity link function, the final predicted values were back transformed via exponentiation (please see equations (3) and (6) of the revised

manuscript). To ensure more clarity as raised by the Reviewer, we have added some texts under the 'TSBHM' approach in the revised manuscript to say:

"Note that the building intensity variable is discrete and could also be modelled using discrete count probability distribution such as Poisson. However, for our purposes, we have used a Normal distribution to model the log-transformed version of the data which was continuous and normally distributed (based on preliminary explorations)."

Additionally, we have added texts within the manuscript under 'Step Two' to highlight that the population density described in the manuscript was defined as people per building, however, the methodology still applies when population density is defined as people per unit settled area:

"Note that while the methodology described here utilised population density D_i defined as people per building, it could readily be applied to population density defined as people per unit settled area."

3. Suppl. Mat., Table S1c, p. 7. Thanks for adding this table, it's very useful. However, Bhat, Chat, Dhat are all described as "Data", but if they're predicted values from a model they can't be observed, so "Data" doesn't seem right. Can this be corrected?

RESPONSE: Thank you very much. Yes, these have now been corrected in Table S1c p.7 and changed from 'Data' to 'Model prediction' in line with the reviewer's important observations.

References

1. Leasure, D. R., Jochem, W. C., Weber, E. M., Seaman, V. & Tatem, A. J. National population mapping from sparse survey data: A hierarchical Bayesian modeling framework to account for uncertainty. *Proc Natl Acad Sci U S A* 117, (2020).
2. Boo, G. *et al.* High-resolution population estimation using household survey data and building footprints. *Nat Commun* 13, (2022).
3. Wardrop, N. A. *et al.* Spatially disaggregated population estimates in the absence of national population and housing census data. *Proceedings of the National Academy of Sciences of the United States of America* vol. 115 Preprint at <https://doi.org/10.1073/pnas.1715305115> (2018).

NCOMMS-24-56010.

Comments for Authors

VERSION 0

Major Points

1. BHM & TSBHM:

- 1.1. I found it hard to keep track of which quantities are directly observed (i.e., the "data") and which are unobserved (i.e., model parameters). E.g., I think the B_i in eqn (1) are the building intensity observations from Planet, while \bar{B}_i is a model parameter (unobserved). Similarly, I assume the C_i are observed population counts from the survey data (but then what are Y_i ?). It might be clearer to use, say, roman letters for data and greek letters for unobserved model parameters. Or capital vs lower case, or something else systematic. A summary table of symbols explaining what each is and whether it is directly observed or not, fixed or random, would be helpful.
- 1.2. You list two sources of population estimates from surveys: urban structural listings and malaria surveys. How did the two different sets of counts enter the model? Is there a model parameter for survey type to, for example, account for any differences in accuracy by survey type?
- 1.3. How does the building intensity model (Step 1) actually correct the bias due to tree cover? It seems this must always be negative as tree cover obscures some structures so they are undercounted. Is tree cover a covariate in the model? Shouldn't only those cells that are under trees should be corrected? As written, it looks like the bias-corrected building intensities will be averages over intensities from both tree covered and non-tree covered mesh nodes, within observed combinations of covariates. This would under-correct tree-covered location but over-correct uncovered locations.
- 1.4. How is the uncertainty about B_i from Step 1 propagated through to Step 2? As written it looks like the point estimates (without uncertainty) from step 1 are just plugged in to step 2.

- 1.5. Step 2: "with appropriate priors (that is, priors that will not lead to model impropriety)". These priors need to be described somewhere.
2. Geospatial covariates:
 - 2.1. If Table S1 has the "final model covariates", what was the initial candidate set?
 - 2.2. Was the stepwise selection done using the 'raw' data (i.e., the data subject to bias due to tree-cover, etc.)? For the final model, was the procedure the same as described in the simulation study?
 - 2.3. Were the same covariates used for the one-step and two-step models? This could be problematic if a particular covariate pattern implies contradictory outcomes in the two steps. Are the beta coefficients at least different between the two steps?
3. Simulation Study (Appendix):
 - 3.1. The point of the simulation study was not clear until the end of the paper, especially the intent to compare the BHM and TSBHM results. You could make this clearer earlier on.
 - 3.2. You state "simulation parameter values were chosen to obtain approximately the same total population count as in the real data model", which I assume means the estimated population from the full data set. But then you multiply the survey and building counts by 20, 30, ... per cent. Did you then expect to recover the original total population count, or the proportionately reduced counts? The sentence "the 'true' population is taken as 11,643,074 people and model performances are adjudged by high close the total population estimates are to the 'true' value across the various proportions of missingness" suggests the former. Put another way, are the population totals in Table S3 inputs to the simulation study or model outputs?
 - 3.3. Step 2: The spatial covariates, x_1, \dots, x_5 are resampled at each iteration. There are twenty of these in Table S1 but the x_k sequence only goes up to 5. What about the rest? Moreover, these look like they're probably correlated from their descriptions (e.g., distance to local roads and distance to main roads). If you just simulate these independently how can you be sure you end up with combinations that make physical sense?
 - 3.4. Step 7: What does "posterior simulation of the results" entail?

- 3.5. How are the coverages p and b applied in the simulation? These symbols don't appear anywhere in steps 1--7.
- 3.6. What are the plotted points in Figure S6? What do the fitted lines represent?
4. Papua New Guinea Application:
 - 4.1. Figure 6: What's the explanation for the systematically higher estimates from the BHM model? Is it that the BHM underestimates B_i because buildings under tree cover are not counted?
 - 4.2. How do your estimates of the total PNG population compare with official estimates from the NSO and international organizations?

Other Points

1. p.10: In Figure 2, Step 2, you have $Y_i \sim \text{Poisson}(D_i * B_i)$, then below you have $D_i = Y_i / B_i$. This would imply that $Y_i \sim \text{Poisson}(Y_i)$ which doesn't make sense. Is something wrong here? In the caption you also have $Y_i = C_i$. Why the two symbols? If one is 'data' and the other is a model parameter it might be useful to make this clearer.
2. p.11, 1st para: You have "the resampled Planet settlement data was resampled". How many times was it resampled?
3. p.11, 1st para: "The BHM scheme assumes that the input settlement building intensity B_i is perfectly observed". I assume this is the single-stage model and that you're comparing it to the two-stage. Maybe make this a bit clearer since both your models are technically BHMs; e.g., add a citation reference to the "old" BHM you're improving.